



# 1 Enhanced Evaluation of Sub-daily and Daily Extreme
# 2 Precipitation in Norway from Convection-Permitting Models at
# 3 Regional and Local Scales

Kun Xie[1,2], Lu Li[3], Hua Chen[1,2], Stephanie Mayer[3], Andreas Dobler[4], Chong-Yu Xu[5], Ozan Mert
Gokturk[3]
[1]State Key Laboratory of Water Resources and Hydropower Engineering Science, Wuhan University, Wuhan
430072, P. R. China
[2]Hubei Provincial Key Lab of Water System Science for Sponge City Construction, Wuhan University, Wuhan,
China
[3]NORCE Norwegian Research Centre, Bjerknes Centre for Climate Research, Bergen, Norway
[4]The Norwegian Meteorological Institute, Oslo, Norway
[5]Department of Geosciences, University of Oslo, P.O Box 1047 Blindern, 0316 Oslo, Norway
*Correspondence to*: Hua Chen (chua@whu.edu.cn); Lu Li (luli@norceresearch.no)
**Abstract**
Convection-permitting regional climate models (CPRCMs) have demonstrated enhanced capability in capturing
extreme precipitation compared to regional climate models (RCMs) with convection-parameterization schemes.
Despite this, a comprehensive understanding of their added values in daily or sub-daily extremes, especially at local
scale, remains limited. In this study, we conduct a thorough comparison of daily and sub-daily extreme precipitation
from HARMONIE-Climate model, cycle 38 at 3km resolution (HCLIM3) and 12km resolution (HCLIM12) across
Norway's diverse landscape, divided into eight regions, using both gridded and in-situ observations. Our main focus
is to investigate the added value of CPRCMs (i.e., HCLIM3) compared to RCMs (i.e., HCLIM12) for extreme
precipitation at daily and sub-daily scales from regional to local scales, and quantify to what extend CPRCM can
reproduce the orographic effect on extreme precipitation at daily and sub-daily scale. We find that HCLIM3 better
matches observations than HCLIM12 for daily and sub-daily precipitation extreme indices at regional scale in
Norway. More specifically, HCLIM3 better captures the maximum 1-day precipitation (Rx1d) at most of the regions
except south-western region in Norway. Notably, HCLIM12 shows underestimation in the complex orography for
annual Rx1d. For the maximum 1-hour precipitation (Rx1h), the superiority from HCLIM3 have also been found on
average, although with slightly higher wet-bias in the western, middle-inland and middle-coastal during summer. In
addition, the reverse orography effect on seasonal Rx1h at regional scale can be better reproduced by HCLIM3 than



HCLIM12 in most seasons except spring. At the local scale, HCLIM3 can better capture the temporal evolution of
Rx1h than HCLIM12 when compared with observations between 1999-2018. However, we see that the benefit from
HCLIM3 in capturing seasonal Rx1d within western region diminishes at local scale. Most interesting finding is that
the added value from HCLIM3 is clearer in Rx1h than in Rx1d at both regional and local scale, especially in the
extreme seasonality. In general, HCLIM3 performs better than HCLIM12 on Rx1d and Rx1h in Norway with the
mean of bias distribution closer to zero, although it varies a bit among regions. Specifically, HCLIM3 performs
slightly poorer in the south-western region. This study highlights the importance of more realistic convection-
permitting regional climate simulations in providing reliable insights into the characteristics of precipitation
extremes across Norway's eight regions. Such information is crucial for effective adaptation management to mitigate
severe hydro-meteorological hazards, especially for the local extremes.
**1 Introduction**
In recent years, the world has witnessed a surge in both frequency and intensity of floods primarily attributed to the
increasing occurrence of intensive rainfall events (Tabari, 2020). These changes underscore the pressing need to
develop a predictive understanding of precipitation extremes for the upcoming decades, given the ongoing globe
warming. The intensification of precipitation extremes under the influence of global warming has the potential to
trigger severe natural hazards and exert significant socioeconomic impacts (Thackeray et al., 2022), which has
gained substantial attention in recent research endeavors. However, most of the previous research in this domain
have been based on the utilization of coarse resolution GCMs with grid sizes exceeding 100 km which have fallen
short in accurately simulating extreme precipitation events and its frequency due to their coarser resolution (Piani et
al., 2010; Wang et al., 2017). Notably, these GCMs tend to produce the largest errors in predicting extreme
precipitation, particularly in cases involving heavier convective activity, as observed in the study by Gervais et al.
(2014a). Despite various bias-correction techniques are applied to mitigate these discrepancies on the GCMs, as well
as employing them as forcing data for regional climate models (RCMs) with grid size larger than 10 km, it remains a
persistent challenge to eliminate the transfer of biases from GCMs to RCMs, as noted by studies such as
Pontoppidan et al. (2018) and Kim et al. (2020). The large resolution gap between GCMs or RCMs and localized
precipitation extremes further constrains the robust simulations of extreme precipitation as highlighted by Li et al.
(2020a). In addition, the reliance on parameterization schemes to represent convection in these coarse resolution
models introduces a significant source of uncertainty in modelling errors (Prein et al., 2015; Kendon et al., 2019).
More frequent and intense precipitation events under global warming stimulate interest in higher resolution and
physics-based models to improve the estimates of short-duration extremes.

Convection-permitting regional climate models (CPRCMs), with grid size of less than 4 km, offer a promising

alternative, which explicitly represent convection, eliminating the need for parameterizations of atmospheric deep
convection. The potential in resolving deep convection and local extremes from CPRCMs lead to the realistic
representation in daily and sub-daily precipitation features, including diurnal cycle, intensity and frequency of heavy
precipitation events, seasonality, spatial-temporal pattern, wet-spell and dry-spell. For instance, CPRCMs have been



proven to reduce the bias and enhance the representation in precipitation intensity and intensity in the Tibetan
Plateau, the highest highland in the world, as shown in Li et al. (2021). In addition to their capability in capturing
precipitation, Liu et al. (2017) also demonstrated the confidence of CPRCMs in estimating snowfall and snowpack
in the central U.S. Furthermore, the importance of CPRCMs in representing dry spell, dry and wet extremes induced
by local convective activity across Africa has also been found in Kendon et al. (2019), Chapman et al. (2023) further
confirmed its benefit in capturing rare rainfall extreme and local feature. In UK, Kendon et al. (2023) and Kent et al.
(2022) have found the benefit of CPRCMs compared to RCMs with convection parameterization schemes.
Additionally, the superior performance in capturing hourly and daily extreme precipitation including return-level,
frequency and intensity from CPRCMs over Alpine in Europe, has also been highlighted by Adinolfi et al. (2021),
Dallan et al. (2023) and Giordani et al. (2023).
Northern Europe has been reported to experience the most increase in precipitation, as indicated by Dyrrdal et
al., (2023), where a novel CPRCMs have been developed within the Nordic Convection Permitting Climate
Projections project (NorCP) based on the convection-permitting HARMONIE-Climate model, cycle 38 (HCLIM38)
at a resolution of 3 km (HCLIM3) and 12km (HCLIM12). Through comparisons of seasonal precipitation, daily
mean precipitation, higher-intensity daily precipitation, the diurnal of hourly precipitation including frequency and
intensity from HCLIM3 and HCLIM12 over Fenno-Scandinavia, Lind et al. (2020) emphasized the add-value of
CPRCM in reproducing extreme precipitation, primarily over complex terrain, compared to coarser-scale model.
Médus et al. (2022) also noted that the summer diurnal cycle of frequency and intensity of hourly precipitation was
correctly captured in HCLIM3 compared to HCLIM12 in the Nordic region, with HCLIM12 underestimating the
diurnal cycle. However, the evaluation and conclusions from Lind et al. (2020) and Médus et al. (2022) were mainly
focused on the large regional and country scale of Fenno-Scandinavia, overlooking the added values of CPRCM at
local scale. Furthermore, Thomassen et al. (2023) observed that HCLIM3 tends to exhibit underestimations in
monthly precipitation and a later evening peak compared to sub-kilometre models. They found that the advantages
of sub-kilometer models were not outstanding. These evaluations were based on gridded datasets, which introduce
uncertainty at the local scale, especially over complex orography (Lussana et al., 2019). As Chapman et al. (2023)
demonstrated, who underscored the importance of assessing rare extreme rainfall events in eastern African using
convection-permitting models and parameterization convection models at both grid and station scales, that the
extreme from grids representing rainfall averaged over a larger area are damped and hence the return-level will be
smaller than observation. They found that the station-derived shape parameters and return levels are aligned with
observations, and suggested the significance of site-specific analysis and evaluations. The error induced by station
density in gridded dataset has also been indicated in Gervais et al. (2014b), who suggested the source of large errors
in gridded dataset when station density is low. Consequently, a comprehensive evaluation and analysis of the added
value from CPRCMs compared with RCMs that incorporates both regional and local scales is crucial for extreme
precipitations.
We acknowledge that Norway, a Nordic country, is representative of diverse climate features due to its
extended latitude, rugged coastline, plateaus, and complex orography. The dominances of precipitation between the



coastal and inland regions over Norway are distinctly different, and most of studies focusing on the hydrology and
meteorology over Norway were based on the divided regions (Vormoor et al., 2016; Poujol et al., 2021; Konstali
and Sorteberg, 2022). By dividing region with its characteristics, a more thorough comprehension of add-value of
CPRCM in capturing extreme precipitation can be reached. Therefore, reliable evaluation about analyzing the add-
value of CPRCM in capturing extreme precipitation should be scaled to region or local scale.

In the complex mountain areas, extreme precipitation is triggered by the interaction of large-scale atmospheric
activity and local orography property, which may cause severe hydrometeorological hazards, such as flash flooding.
However, understanding the orographic impact in precipitation in complex orography is challenging due to sparse
observations (Rossi et al., 2020). The poor representation of RCMs in capturing local precipitation have been
indicated in Knist et al. (2020). Importantly, CPRCMs shows superior in reproducing precipitation bias over higher
complex orography in the Alpine, as shown in Lind et al. (2016) and Reder at al. (2020). Furthermore, the better
representation of sub-daily and daily heavy precipitation from CPRCMs over the Alpine have also been found in
Ban et al. (2020) and Dallan et al. (2023). Marra et al. (2021) and Dallan et al. (2023) also confirmed the efficiency
of CPRCMs in reproducing the reverse orography effect on hourly extreme precipitation. Conversely, Rossi et al.
(2020) and Mahoney et al. (2015) found the weak depend of subdaily precipitation on elevation in the Colorado,
USA. Opposing the orographic enhance on daily precipitation, Dallan et al. (2023) indicated the no evident relation
of daily precipitation on elevation. It is worth noting that these ambiguous results were based on the annual maxima,
and the added value from CPRCMs than RCMs have not been explored.  Moreover, the dependence on seasonality
for the performance of CPRCMs especially in complex orography need the evaluation based on season. Thus, we fill
this knowledge gap by characterizing orographic impact on hourly and daily extreme precipitation seasonally.

As highlighted by Konstali and Sorteberg (2022),  there can be significant uncertainties associated with the
interpolation of grided precipitation data. Besides, the benefit for precipitation spatial evaluation based on in-situ
observation has also been reported in Thomassen et al. (2023). Therefore, the evaluation of extreme precipitation
from HCLIM3 and HCLIM12 here, is based on both gridded precipitation and in-situ observation. Our study aims to
address the value of CPRCM (HCLIM3) in capturing the characteristics of extreme precipitation in Norway,
comparing it with a coarser resolution model (HCLIM12) as well as both of the in-situ and gridded precipitation
observations. Here, our contribution to the existing literature, e.g., Médus et al. (2022), revolves around the added
value of CPRCM in the extreme precipitation characteristics, encompassing a range of metrics.

The main objectives of this study are (1) enhance understanding of convection-permitting climate models by
comparing their effectiveness in simulating extreme precipitation with that of regional climate models from regional
to local scales, highlighting the added value of CPRCMs; (2) assess HCLIM3's capability in depicting orographic
effects on seasonal extreme precipitation. This research explores whether the benefits provided by CPRCMs hold
consistently in different regions driven by varying physical processes for precipitation. Finally, our study delves into
the analysis of the intensity and frequency of extreme precipitation events, offering insights into local and regional
variations.





## 2 Study area and data

### 2.1 Study area

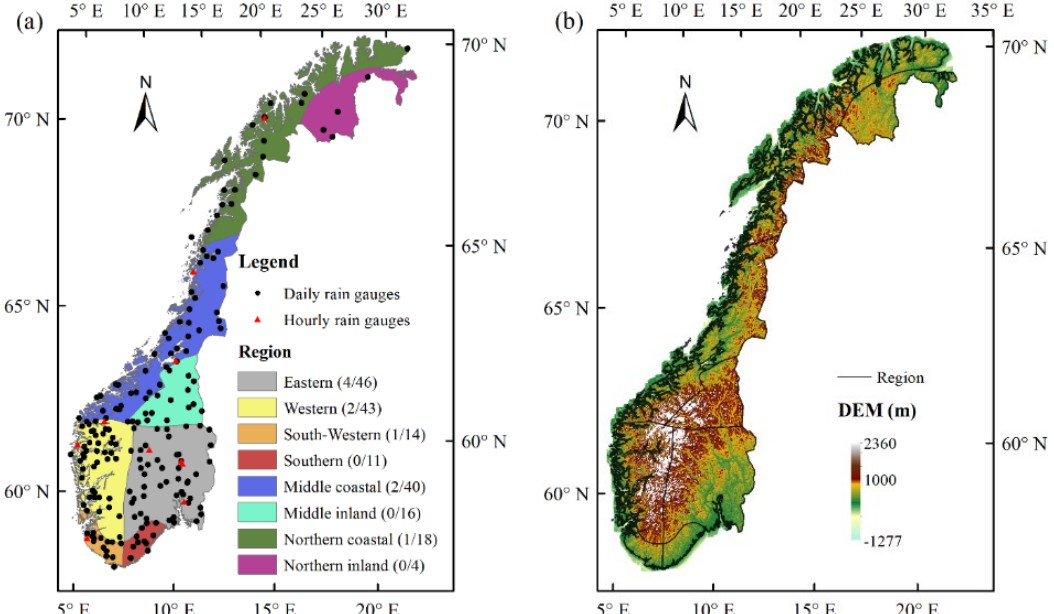

**Figure 1: (a) The division of 8 precipitation regions in Norway. In the legend, the numbers shown in the brackets after each region represent the available size of hourly / daily stations in the region during 1999 – 2018. For example, Eastern (4/46) means that there are available data from 4 hourly stations and 46 daily stations in the Eastern region during 1999-2018; (b) Spatial distribution of topography over Norway.**

The different climate regimes between coastal and inland regions over Norway compels the analysis of hydro-meteorology based on divided regions. We divided Norway into eight regions according to similar season cycle characteristics (Michel et al., 2021): Eastern (E), Southern (S), South-Western (SW), Western (W), Middle-Inland (MI), Middle-Coastal (MCo), Northern-Coastal (NCo), and Northern-Inland (NI), shown in Fig.1.

The study areas cover the mainland Norway which has unique climate characteristics within different regions. The eastern region with stratiform precipitation originating from south and south-east is dominant by continental climate, with convective precipitation in summer. Most of extreme events occurring in the western region with abrupt topography are mainly related to atmospheric rivers (AR), peaking in winter. The wettest region strongest affected by the North Atlantic storm track with enhanced precipitation from front systems and land-falling storms due to the uplift over the Scandes (Poujol et al., 2021), is the west coast of Norway. For the middle-coastal and northern coastal regions, 59% of extremes are associated with AR, and the precipitation rate decreases moving inland (Konstali and Sorteberg, 2022). Northern-inland and middle-inland are the driest regions with lower wet-day intensity (<6 mm/day) and wet-day frequency (<33%), most extremes during summer are linked to AR for the





northern-inland where largest precipitation is dominant by cyclones. South-western region lies at the end of the
climatological jet and is regularly hit by ARs especially during the Zonal and Atlantic Trough weather regimes
(Michel et al., 2021). The main precipitation in Norway is winter and autumn precipitation. These precipitation
patterns in spatial and seasonality are mainly linked to ARs (Schaller et al., 2020; Benedict et al., 2019).

**2.2 Data**
We utilize the outputs of double nesting from HCLIM38 model based on the ALADIN-HIRLAM NWP system,
which include different configuration settings for each spatial resolution: HCLIM3 and HCLIM12 with hydrostatic
dynamics. HCLIM12 covers over most part of Europe with 313 ×349 grid-points using the ERA-Interim reanalysis
~80 km as boundary condition for every 6 h, and HCLIM3 spans over the Fenno-Scandinavia region with 637×853
horizontal grid-points using the HCLIM12 as boundary condition for every 3 h. Importantly, the convection-
parameterization scheme was switched-off in HCLIM3, allowing for an explicit representation of convection
processes. The present-day simulations from HCLIM3 and HCLIM12 span over the years 1997-2018. For more
comprehensive information, refer to the work of Lind et al. (2020) and Médus et al. (2022).
This study primarily centers on assessing the performance of HCLIM3 and HCLIM12 in simulating sub-daily
and daily extreme precipitation events in the present-day (1999-2018) in Norway mainland. The models' outputs are
specifically extracted for Norway mainland. Before analysis, HCLIM3 data was resample to the HCLIM12 grid
(12km) using a bilinear method.
seNorge2018 (SeNorge) covering Norway with 1-day temporal and 1 km spatial resolution (Lussana et al.,
2019) is used as the observation to evaluate the performance of HCLIM3 and HCLIM12 during 1999-2018.
SeNorge2 with 1-hour and 1 km spatial resolution is also applied to evaluate the hourly result during 2010-2018
(Lussana et al., 2018). In addition, in-situ observations including hourly and daily resolution are downloaded from
Norwegian Meteorological Institute Frost API (met.no).
**3 Methods**
**3.1 Evaluation of precipitation**
To evaluate the characteristics of precipitation extremes between HCLIM3 and HCLIM12, we compared the
historical simulations with daily SeNorge gridded dataset, hourly SeNorge2 gridded dataset and in-situ observations.
We only keep the stations that have less than 10% of the data missing during 1999-2018 and consider station
distribution uniformity, which give a total of 192 daily stations and 10 hourly stations, respectively, over Norway
(Fig. 1 and Table 1). In this study, the evaluation based on in-situ observation and gridded dataset (SeNorge and
Senorge2) was defined as the local scale and regional scale, respectively.
For the evaluation at regional scale, HCLIM3 and SeNorge, SeNorge2 were resampled to HCLIM12 grid~12
km. Therefore, the observed and simulated extreme indices were calculated at grid scale, and then averaged the
extreme indices within the corresponding region. For the evaluation based on in-situ observation, HCLIM3 and


HCLIM12 were interpolated to the 192 daily rain-gauges and 10 hourly rain-gauges to calculate the indices using
bilinear interpolation.
For the evaluation of present-day extreme precipitation, we examined the maximum 1-day precipitation
(Rx1d), maximum 1-hour precipitation (Rx1h), return-period-based precipitation amounts at 5, 10, 20, and 50-year
return periods, frequency of daily precipitation exceeding 10, 15, 20 mm, and seasonality of frequency/intensity
from regional to local scales. The calculation of seasonal Rx1d/Rx1h was based on the maximum value within one
season per year.

**Table 1. The information for the ten hourly rain gauges.**

| Name | Station ID | Longitude (°E) | Latitude (°N) | Elevation | Region |
|---|---|---|---|---|---|
| Østre Toten - Apelsvoll | SN11500 | 10.8695 | 60.7002 | 264 | Eastern |
| Ås - Rustadskogen | SN17870 | 10.8107 | 59.6703 | 120 | Eastern |
| Kise in Hedmark | SN12550 | 10.8055 | 60.7733 | 128 | Eastern |
| The Onion of Volbu | SN23500 | 9.063 | 61.122 | 521 | Eastern |
| Kvithamar | SN69150 | 10.8795 | 63.4882 | 27 | Middle-Coastal |
| Tjøtta | SN76530 | 12.4255 | 65.8295 | 21 | Middle-Coastal |
| Stryn - The Hook | SN58900 | 6.5585 | 61.9157 | 208 | Western |
| Fureneset | SN56420 | 5.0443 | 61.2928 | 7 | Western |
| Særheim | SN44300 | 5.6508 | 58.7605 | 87 | South-Western |
| Tromsø - Holt | SN90400 | 18.9095 | 69.6538 | 20 | Northern-Coastal |


**3.2 Extreme precipitation indices**
Generalized extreme value (GEV) distribution function was used to derive the precipitation at specified return
periods (5, 10, 20, 50-year) from the statistical cumulative distribution functions of the conceptual distributions for
the annual maximum precipitation derived from the precipitation series. GEV distribution has been widely used to
model extreme events in meteorology (Coles et al., 2003). The cumulative distribution function $F(x)$ and
probability density function $f(x)$ of GEV were as follows to calculate the return level $Z_p$:

$$F(x) = exp\left\{ -\left[ 1 - k\left( \frac{x - \xi}{\alpha} \right) \right]^{1/k} \right\}, k \neq 0$$

$$f(x) = \frac{1}{\alpha}\left[ 1 - k\left( \frac{x - \xi}{\alpha} \right) \right]^{1/k-1} exp\left\{ -\left[ 1 - k\left( \frac{x - \xi}{\alpha} \right) \right]^{1/k} \right\}$$

$$Z_p = \xi - \frac{\alpha}{k}\{1 - [-log\,(1 - p)]^{-k}\}$$






Where, $\alpha$, $\xi$, and $k$ indicates the scale, location and shape parameter, respectively. Kolmogorov-Smirnovs,
Anderson-Darlings, and Chi-Square tests were performed to determine if the GEV was accepted to fit the maxima
series.

**3.3 Quantification of the orographic effect**
The orographic effect on Rx1h and Rx1d precipitation was explored by looking at the relationship with elevation of
the annual and seasonal maxima from regional to local scales. A linear regression model (Di Piazza et al., 2011) was
utilized to approximate the relations. The relation of elevation with observation (Rx1h: SeNorge2; Rx1d: SeNorge
and daily in-situ observation) and simulation (HCLIM3 and HCLIM12) was fitted to compute the linear regression
slopes and expressed as an averaged precipitation (mm) per kilometer of elevation. The orographic effect at local
scale was only based on daily in-situ observation due to the limited hourly in-situ observation. At local scale, the
elevation for each rain-gauges was extracted according to the digital elevation model. At regional scale, the grid of
digital elevation model and HCLIM3 was resample to the same grid resolution of 12 km as HCLIM12 before
calculation. Only the grids and stations above sea level of 0 m are included to quantify the orographic effect.

If the precipitation increases with elevation, it means that the orographic effect on extreme precipitation; if the

precipitation decreases with elevation, its means that the reverse orographic effect on extreme precipitation.

**4 Results**
**4.1 Evaluation of daily extreme with SeNorge**
**4.1.1 Rx1d precipitation**

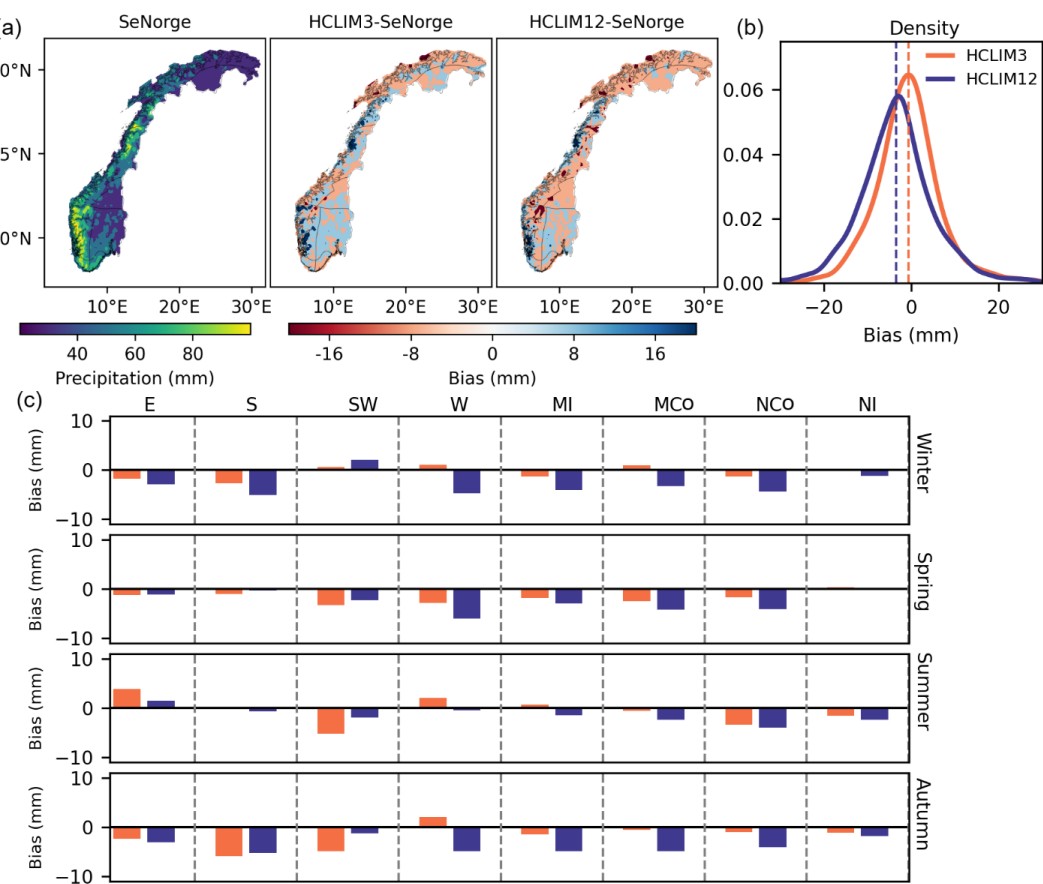


**Figure 2: (a) Comparison of annual maximum 1-day precipitation (Rx1d) during 1999-2018; (b) density plot of the bias**
**distribution from HCLIM3 and HCLIM12 compared to SeNorge for Rx1d during 1999-2018; (c) the bias of seasonal**
**Rx1d from HCLIM3 and HCLIM12 to SeNorge for eight regions. For each region, the result is the bias is the average bias**
**of grids within the region. The absolute differences are equal to model simulations minus observations, divided by**
**observations.**

Figure 2 provides a comprehensive comparison of Rx1d bias from HCLIM3 and HCLMI12 compared to SeNorge.
From Fig. 2 (a), we can see that HCLIM12 has more grids with underestimated Rx1d than HCLMI3 in Norway,
which is confirmed clearly in Fig. 2 (b) showing density plot of the bias distribution from two models compared
with SeNorge. Specifically, more grids from HCLIM3 tend to overestimate Rx1d within the 0-10 mm/day range,
while HCLIM12 leans towards underestimation. In addition, referred to Fig. S1, the spatial distribution of seasonal
Rx1d is also shown. The density curve in Fig. 2 (b) reflects a higher peak at 0 for HCLIM3, indicating a more
accurate representation of Rx1d with an average bias closer to 0. Conversely, HCLIM12 shows a dry-bias for Rx1d
on average. Compared to the bias of annual precipitation in Fig. S2, we noted that HCLIM12 shows higher wet-bias
for annual precipitation. This could be attributed to the overestimation of low-moderate precipitation (drizzle) in





HCLIM12, as suggested by Lind et a. (2020), who also highlighted the higher drizzle in HCLIM12, when comparing
the contribution of intense precipitation to total precipitation.

Figure 2 (c) shows the bias of Rx1d from HCLIM3 and HCLIM12 compared to SeNorge for eight regions in

four seasons. HCLIM3 better captures Rx1d in winter for all regions than HCLIM12, while HCLIM12 shows higher
dry-bias across regions except in the south-western region. In autumn, HCLIM3 shows more underestimation of
Rx1d in the southern and south-western regions but performs better elsewhere. Both HCLIM3 and HCLIM12 have
dry-bias for spring Rx1d in all regions except northern-inland with almost no-bias, and HCLIM12 has more
underestimation than HCLIM3 except in the southern and south-western regions. In summer, HCLIM3 outperforms
Rx1d in the 5 out of 8 over HCLIM12. Overall, HCLIM3 shows notably added value in Rx1d comparing with
HCLMI12 across regions and seasons although with exception of the south-western region.

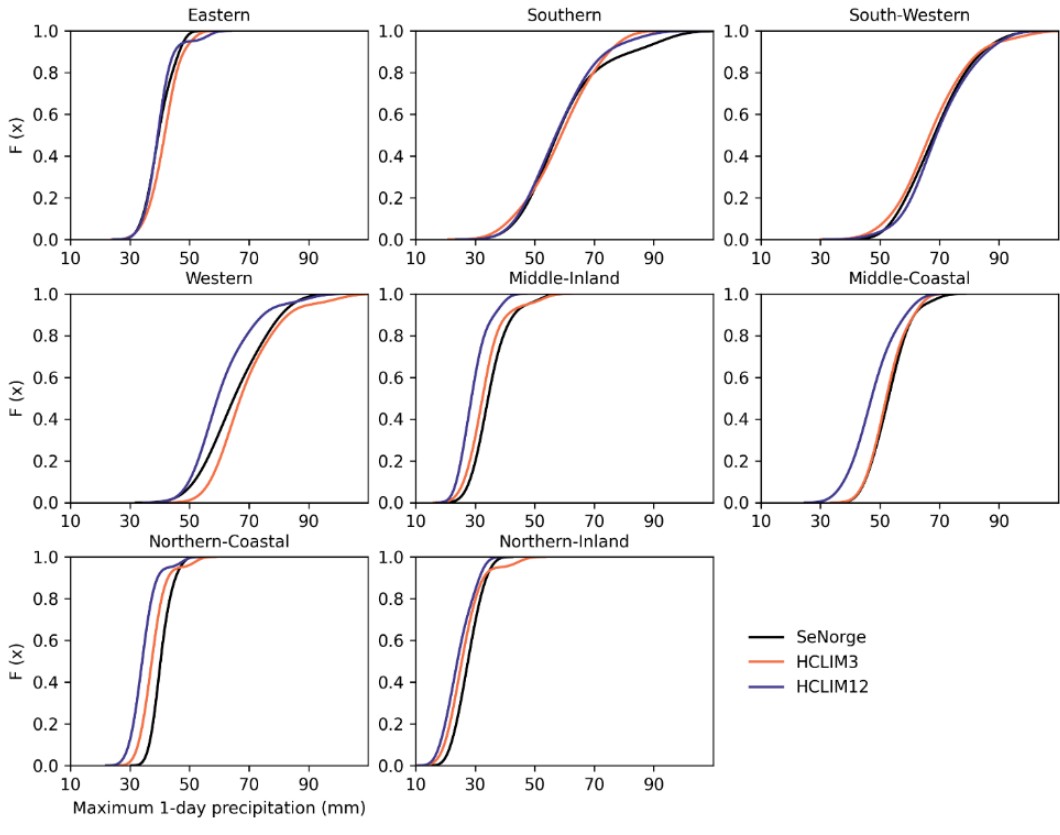


**Figure 3: The empirical distribution of annual Rx1d during 1999-2018 in each of the eight Norway regions from SeNorge,**
**HCLIM3 and HCLIM12.**

Furthermore, Fig. 3 shows the empirical distribution of Rx1d from HCLIM3 and HCLIM12 over all eight

regions during 1999-2018, compared to the SeNorge data. The distribution of Rx1d from HCLIM 3 and HCLIM12
varies among regions with intraregional differences. For example, Rx1d in the western region is overestimated by



HCLIM3, while underestimated by HCLIM12. Besides, both HCLIMs underestimate the Rx1d in the middle-inland,
northern-coastal and northern-inland generally. HCLIM3 demonstrates overall closer alignment with SeNorge in
four regions, specifically the western, middle-inland, middle-coastal and northern-coastal regions, compared to
HCLIM12. In the eastern region, HCLIM3 tends to overestimate Rx1d when it is above 33.5 mm, while exhibits
better performance when Rx1d is larger than 48.2 mm compared with HCLIM12. In contrast, in the northern-inland,
HCLIM3 can better capture the Rx1d except the severe extremes. In the southern and south-western regions, the
distributions from both HCLIMs are quite similar as SeNorge, posing challenges in discerning superiority.
**4.1.2 Return levels**

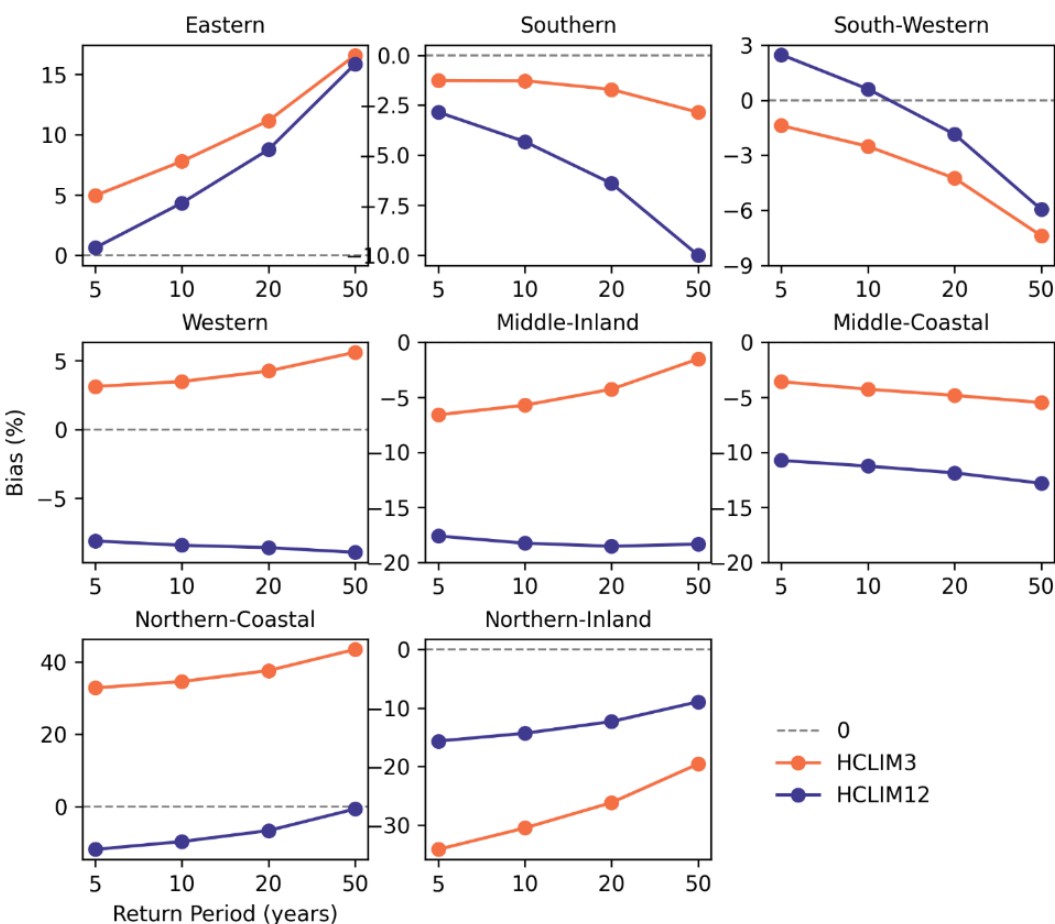


**Figure 4: The bias of extreme annual Rx1d exceeding the 5-year to 50-year over eight regions between seNorge and**
**HCLIMs (i.e., HCLIM3 and HCLIM12).**





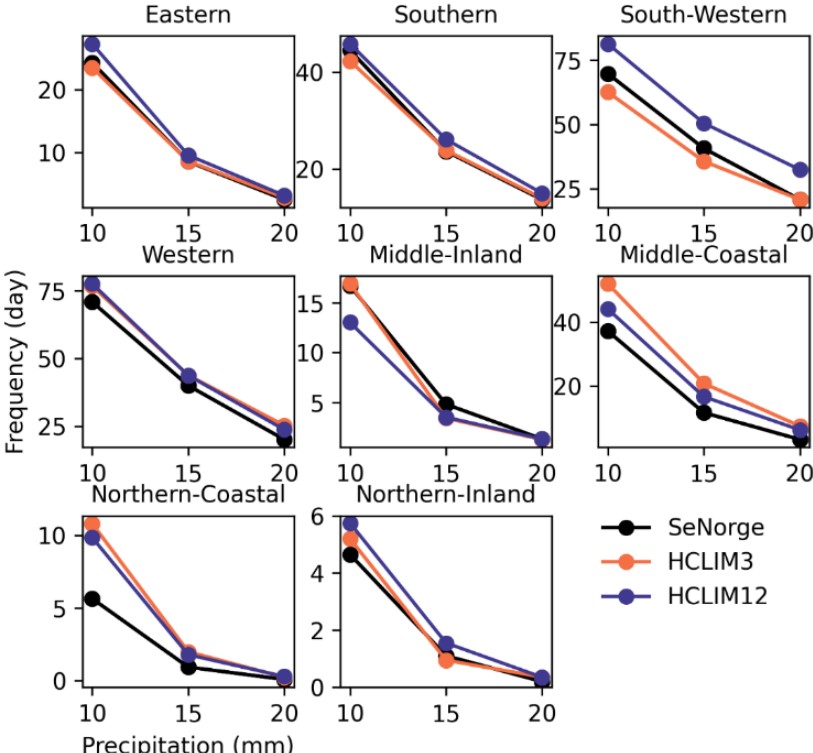


**Figure 5: The frequency of daily extreme precipitation exceeding 10, 15, 20mm/day.**


Figure 4 shows the bias in estimated daily precipitation for 5-, 10-, 20-, and 50-year return periods during 1999-2018
across eight regions (compared to SeNorge). The great interregional variation is shown between HCLIM3 and
HCLIM12. Relative to SeNorge, HCLIM3 tends to overestimate return-levels in the eastern, western and northern-
coastal regions, while underestimates them in the others. With the exception of the western and northern-coastal
regions, HCLIM12 shows a bias direction similar to HCLIM3. The performance of HCLIMs in capturing extremes
varies across regions. HCLIM3 consistently outperforms HCLIM12 in the southern, western, middle-inland and
middle-coastal regions for all return-periods, but performs less satisfactorily in other regions. As the return period
increases, the bias between HCLIMs and observations increases, except in the northern-inland region.
The frequency of precipitation exceeding 10, 15 and 20 mm is compared between simulations and observations
for eight regions in Fig. 5. The south-western and western regions experience frequent extreme precipitation events
exceeding 10, 15, and 20 mm/day. With exception in the western, middle-coastal and northern-coastal regions,
HCLIM3 can better capture the frequency of extreme precipitation than HCLIM12. HCLIM12 tends to overestimate
the frequency of extreme precipitation in most regions, except the middle-inland region. Both HCLIM3 and
HCLIM12 well capture the frequency as the severity of extremes increases.



Given the societal impacts of precipitation extremes, understanding how HCLIM3 and HCLIM12 represent
these extremes is crucial. The physical processes driving precipitation in inland and coastal regions, as highlighted
by Konstali and Sorteberg (2022), emphasize the need for a separate evaluation for each region with different
characteristics.  This approach ensures a more robust assessment, providing valuable information for regional
authorities.
**4.1.3 Evaluation of hourly extreme with SeNorge2**

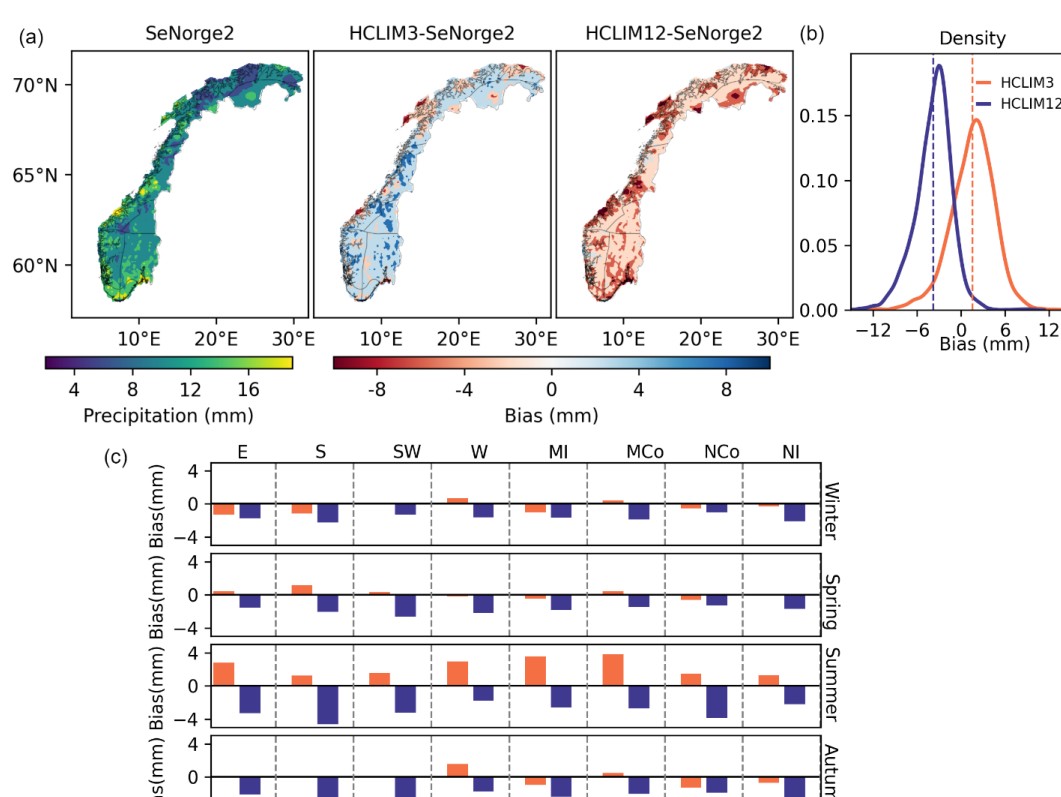

**Figure 6: (a) Comparison of annual Rx1h during 2010-2018; (b) density plot of bias distribution from HCLIM3 and**
**HCLIM12 compared to SeNorge2 during 2010-2018; (c) the bias of seasonal Rx1h from HCLIM3 and HCLIM12 to**
**SeNorge2 for eight regions. For each region, the result is the bias is the average bias of grids within the region.**

Figure 6 provides a comprehensive comparison of Rx1h bias from HCLIM3 and HCLMI12 comparing with
SeNorge2 during 2010-2018. From Fig. 6 (a), we can see that HCLM3 overestimate the annual Rx1h, while
HCLIM12 underestimate the Rx1h almost over Norway. Furthermore, the density plot of bias in Fig. 6 (b) showing
the bias distribution from two models compared with SeNorge2, further confirmed the overestimation from
HCLIM3 and underestimation from HCLIM12. On average, the annual Rx1h in most grids could be better captured





by HCLIM3 with an average bias (1.5 mm) closer to 0, than HCLIM12.  Conversely, HCLIM12 underestimate Rx1h
about 4 mm on average.

The bias of Rx1h from HCLIM3 and HCLIM12 compared to SeNorge2 for eight regions in four seasons is

shown in Fig. 6 (c). HCLIM3 performs better in capturing seasonal Rx1h than HCLIM12 in most regions except
western, middle-inland and middle-coastal regions during summer. Besides, the seasonal Rx1h except summer is
better represented by HCLIM3 compared to HCLIM12. In summer, wet-bias in all region is observed from
HCLIM3. HCLIM12 shows dry-bias in all regions and seasons.

**4.2 Evaluation of daily extreme with in-situ data**
**4.2.1 Rx1d precipitation**

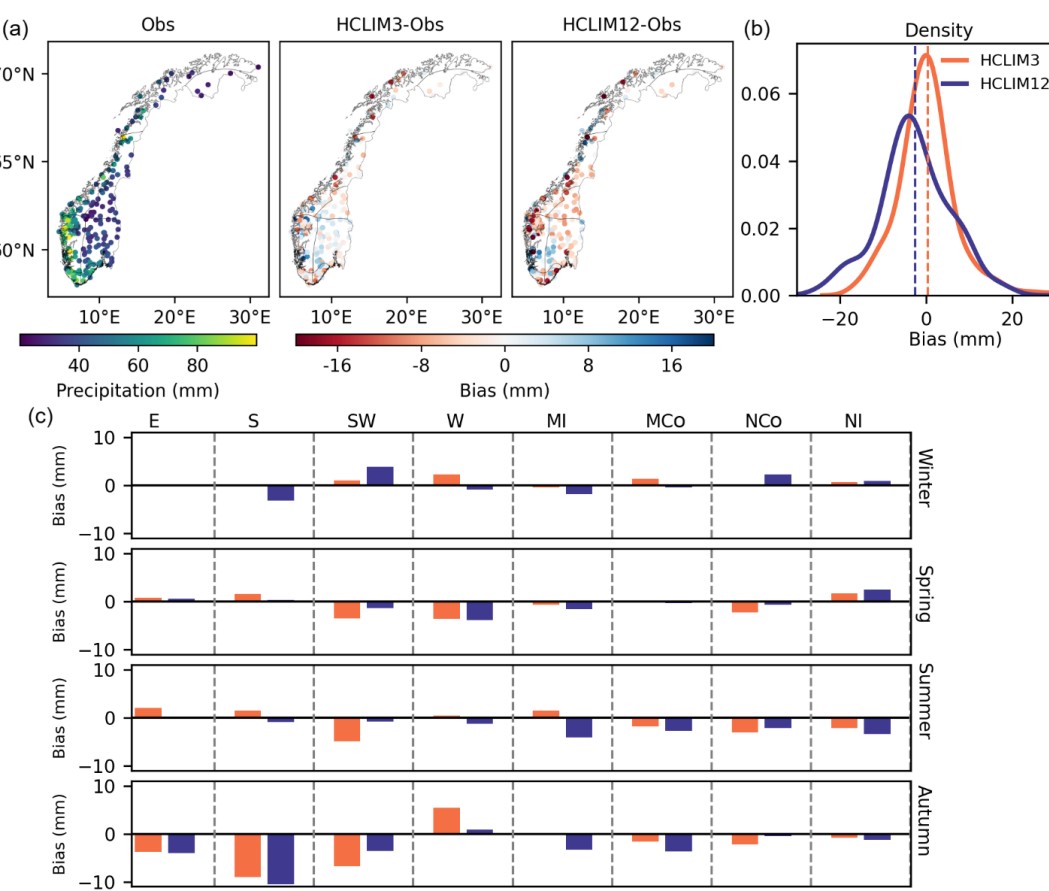


**Figure 7: (a) The annual Rx1d of in-situ observation, and the bias of Rx1d from HCLIM3 and HCLIM12 to in-situ**
**observation during 1999-2018 over 194 stations; (b) density distribution of Rx1d bias between HCLIMs and observations**
**from 194 stations during 1999-2018; (c) the bias of seasonal Rx1d between HCLIMs and observations across the eight**
**regions. For each region, it is the averaged bias from all stations in the region.**





335 Similar to the regional results in Fig. 2, Fig. 7 shows the annual and seasonal bias of Rx1d from HCLIM3 and

336 HCLIM12 in comparison to in-situ observations. Notably, a larger difference between HCLIM3 and HCLIM12 can

337 be seen at local scale compared to regional result: a greater number of grids from HCLIM3 approach zero-bias, and

338 more grids from HCLIM12 shows a 20 mm dry bias, as shown in Fig. 7 (b). On average, HCLIM12 tends to

339 underestimate annual Rx1d, while HCLIM3 represents added value in capturing annual Rx1d at local scale.

340 When examined seasonally, as shown in Fig. 7 (c), excepting the western region with a wet bias, HCLIM3

341 shows better performance in capturing winter Rx1d. In spring, HCLIM12 falters in representing Rx1d in the

342 southern, south-western and northern-coastal regions, while HCLIM3 shows beneficial in most regions except the

343 south-western, western and northern-coastal regions during autumn. Notably, the performance of both HCLIM3 and

344 HCLIM12 at the local scale differs from the regional results. For example, their performance in the southern and

345 south-western regions during autumn deteriorates, displaying a more significant dry bias. In contrast, biases in the

346 eastern region from both HCLIM3 and HCLIM12 decrease in all seasons except winter.

347 Biases from HCLIM3 increase in the western region during winter, spring and autumn. HCLIM12 shows

348 improvement in capturing Rx1d at local scale compared to regional result except southern and south-western

349 regions. However, the benefit from HCLIM3 over HCLIM12 diminishes in the western region at local scale relative

350 to the regional scale. Notably, HCLIM3 fails to capture seasonal Rx1d in the south-western region, which is the

351 same as from the regional results. Therefore, HCLIM3 and HCLIM12 yield divergent outcomes across different

352 regions and seasons when examined at local scale compared to the regional scale.







### 4.2.2 Return-levels


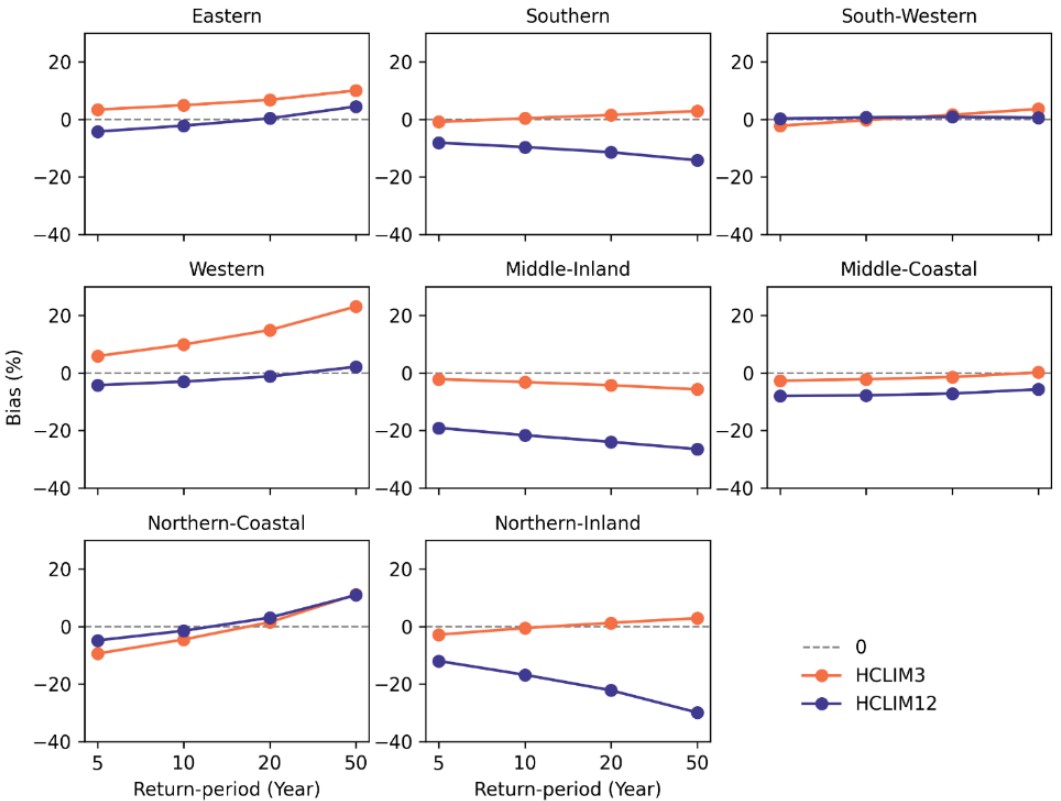


**Figure 8: Percentage biases of daily precipitation in the present-day for HCLIM3 and HCLIM12 relative to observation for 5-, 10-, 20-, and 50-year return periods in the 192 daily rain-gauges. Return periods of 5-, 10-, 20-, and 50-year are calculated on the basis of station-scale GEV. The bias of daily precipitation with different return-levels for each region is the averaged bias of all stations within the corresponding region.**


Figure 8 shows bias in estimating daily return levels (e.g., 5-, 10-, 20-, and 50-year return periods) from HCLIM3
and HCLIM12 compared to observations during the 1999-2018. This figure illustrates the average bias for return-
levels at grids within the corresponding regions. Biases from HCLIM3 and HCLIM12 exhibit variations across
regions and return periods. HCLIM3 has a higher bias in the eastern, western and northern-coastal regions compared
to HCLIM12, similar to the result at regional scale, while HCLIM12 shows larger uncertainty in the eastern region.
Despite exceptions, HCLIM3 generally provides a more accurate representation of the return levels. Notably, in the
northern-coastal, HCLIM3 exhibits lover bias for the return-level under 20-year return-period, but larger bias for the
return-level under 5, 10, and 50-year return-periods. Both HCLIM3 and HCLIM12 perform well in the south-
western region. In addition, the range of the return levels from all grids within the corresponding region is shown in





Fig. S3. In the western and northern-inland regions, HCLIM3 introduces larger uncertainty, as evident by a wider
whisker, in comparison to HCLIM12. Additionally, with longer return periods, both the bias of extremes and its
associated uncertainty estimated by HCLIMs tend to increase.

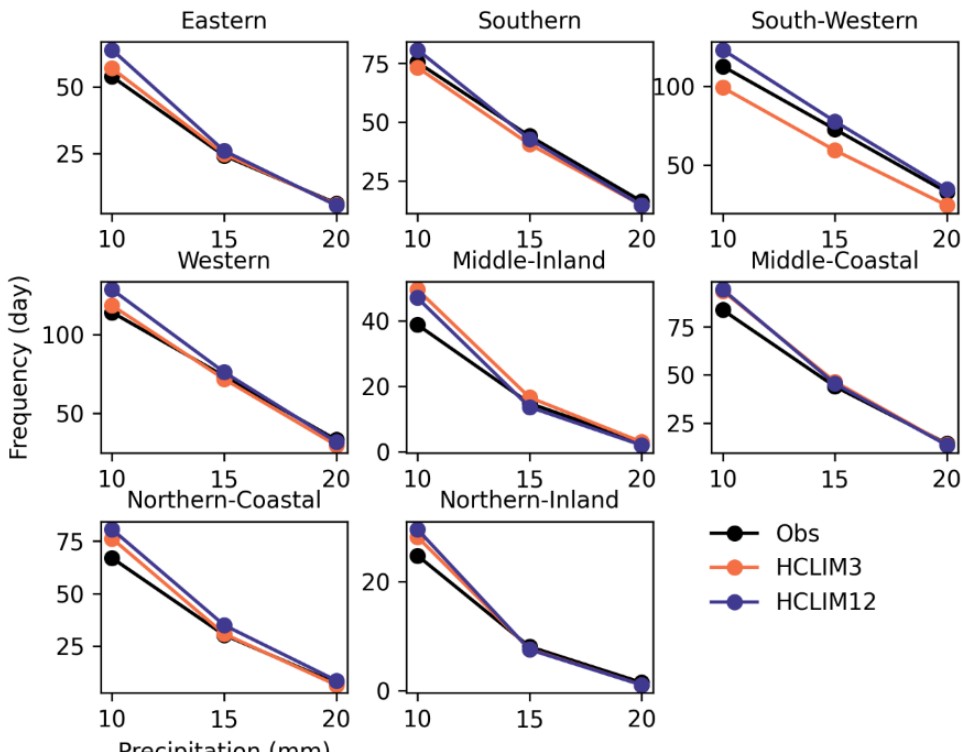


**Figure 9: The frequency of daily precipitation exceeding 10, 15, 20mm/day. The frequency for each region is the averaged**
**frequency of all stations within the corresponding region.**

By comparing the frequency of extreme precipitation exceeding 10, 15, 20 mm/day from HCLIM3 and
HCLIM12 to in-situ observation, as shown in Fig. 9, we find HCLIM3 outperforms HCLIM12 in capturing the
frequency of extreme precipitation in the eight regions generally, except the south-western, middle-inland and
middle-coastal regions.  For the frequency of extreme precipitation above 20 mm/day, both HCLIMs capture it well.
Compared to the result in regional scale (Fig. 5), the frequency of daily extreme precipitation between regional
scale and local scale is different. The frequency at local scale is higher than the regional scale across all regions.
Besides, the intraregional difference for the added value of HCLIM3 is also different. For example, the added value
of HCLIM3 is shown in the middle-coastal according to the regional result, while not shown according to local
result. In general, the benefit of HCLIM3 in capturing the frequency of extreme precipitation is seen both in the
regional and local scale.




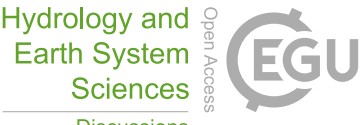

### 4.3 Evaluation of hourly extreme with in-situ data

### 4.3.1 Rx1h intensity

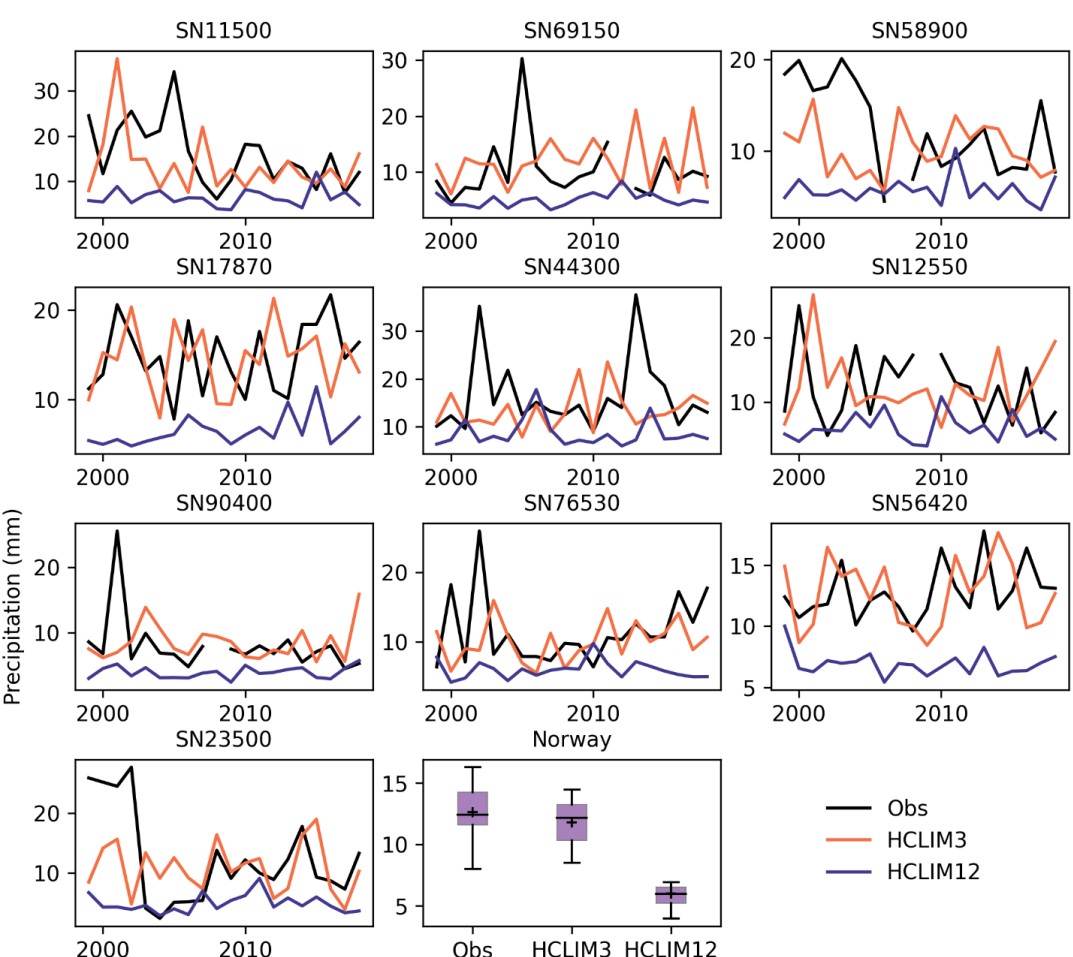

**Figure 10: Time evolution of Rx1h precipitation for each year from observation, HCLIM3 and HCLIM12 during 1999-2018 at 10 rain-gauges.**

The time evolution of annual Rx1d from HCLIM3 and HCLIM12 is compared to in-situ observation during 1999-2018, as shown in Fig. 10. Compared to HCLIM12, HCLIM3 shows distinct superior in capturing the time evolution of annual Rx1d, even with underestimation and time shifting at some local places. For example, the annual Rx1d above 25 mm/day at SN69150, SN44300, SN90400, SN76530 and SN23500 is struggle to captured by HCLIM3 and HCLIM12. However, HCLIM3 well capture the annual Rx1d in other local places, despite of the time deviation of





annual Rx1d. Taking the site SN11500 as an example to illustrate the time deviation: HCLIM3 simulate that the
annual Rx1d (37 mm) in the past 20 years was in 2001, four years earlier than the in-situ observation (35 mm).
Furthermore, to better assess the annual variability of Rx1h, we extracted grids within a 12 km radius of each station
and calculate the uncertainty range (Fig. S4). A comparison between HCLIM3 and HCLIM12 reveals that the
interpolated local Rx1h precipitation from HCLIM12, particularly over grids with a larger area, tends to be damped,
resulting in smaller return levels and a narrower range than HCLIM3. Importantly, the Rx1h in the past 20 years
(1999-2018) can be well captured by HCLIM3. In the view of station statistics for the mean annual Rx1d in Norway
(Fig.10) using boxplot, the mean annual Rx1d from HCLIM3 is within the range of observation, while HCLIM12 is
all below the minimum value of observation. Despite outperforming than HCLIM12, it is noteworthy that HCLIM3
demonstrates limitations in reproducing the accurate occurrence time and magnitude of annual Rx1h at local-level in
Norway.




**4.3.2 Return levels**

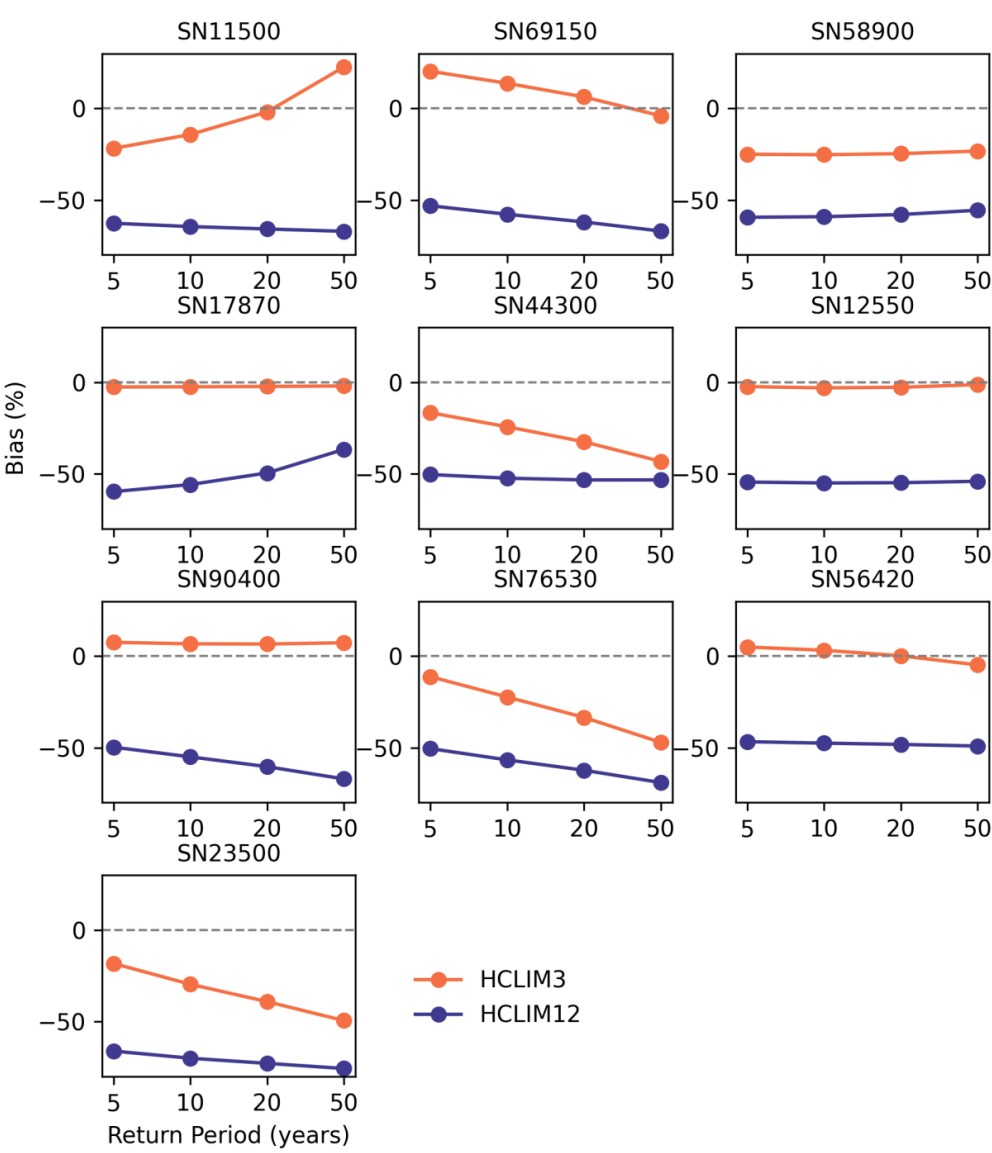


**Figure 11: The bias of Rx1h precipitation exceeds 5-year, 10-year, 20-year, and 50-year return levels between HCLIMs**
**and in-situ observations (based on GEV method).**


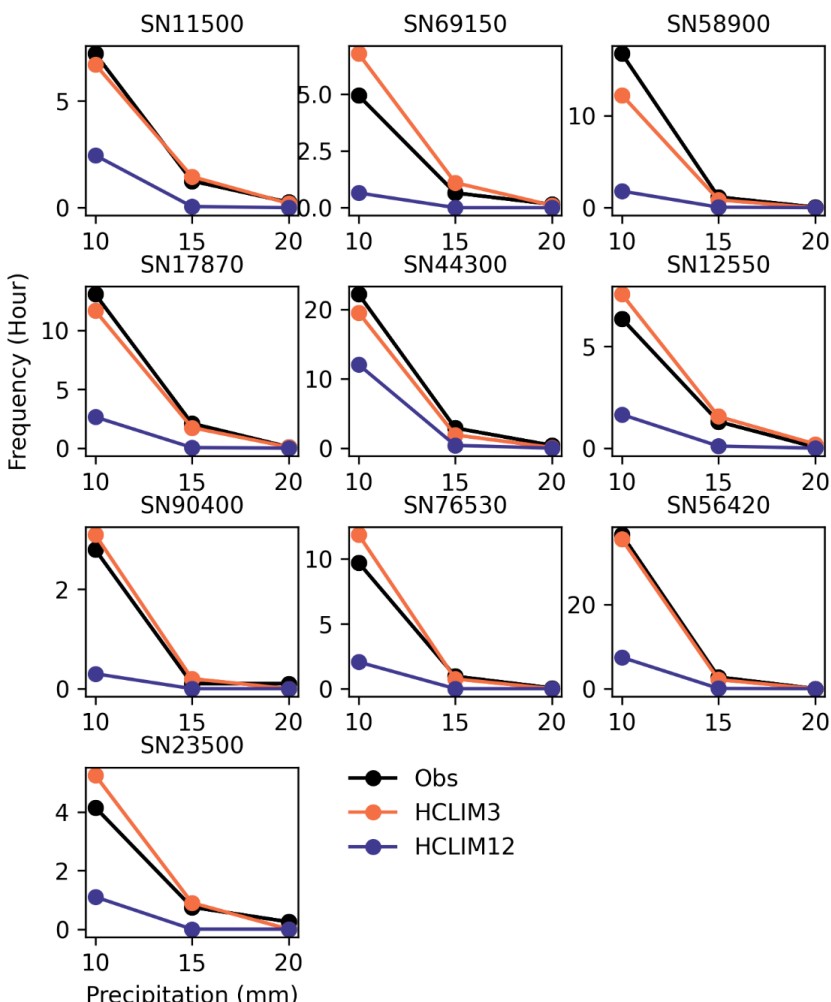

**Figure 12: The frequency of hourly extreme precipitation exceeding 10, 15, 20mm from HCLIM3, HCLIM12 and in-situ observations.**

At the local and hourly scale, HCLIM3 has better representation of the frequency of hourly extreme events compared to HCLIM12. Despite both HCLIM3 and HCLIM12 underestimating the Rx1h precipitation for 5, 10, 20, and 50-year return periods at almost stations (Fig. 11), the bias for ten rain gauges between observations and interpolated HCLIM3 is consistently lower than that from HCLIM12 at all return periods. There is an exception, at SN11500, SN69150, SN90400 and SN56420, HCLIM3 shows slight overestimation. Moreover, the bias between HCLIMs and in-situ observation increased as return period increases. Notably, the return level of hourly extreme events at SN17870, SN12550, SN90400 and SN56420 is accurately captured by HCLIM3, indicating its ability to better capture the extreme hourly precipitation at 5,10,20,50-year return periods compared to HCLIM12 in localized



areas. This result shows the superiority of CPRCM in representing the frequency of extreme precipitation at a very
localized scale, despite the underestimation of return levels from both HCLIM3 and HCLIM12.
In addition, Fig. 12 shows the frequency of hourly precipitation exceeding 10, 15 and 20 mm at ten stations.
The results further confirm the added value of CPRCM in capturing the frequency of extreme precipitation at very
localized scale, despite the tendency to overestimate the frequency. This added value is more obvious than at the
regional scale, although we acknowledge the uncertainty in the extreme precipitation analysis based on the
stationary GEV method.

**4.4 Evaluation of seasonality**

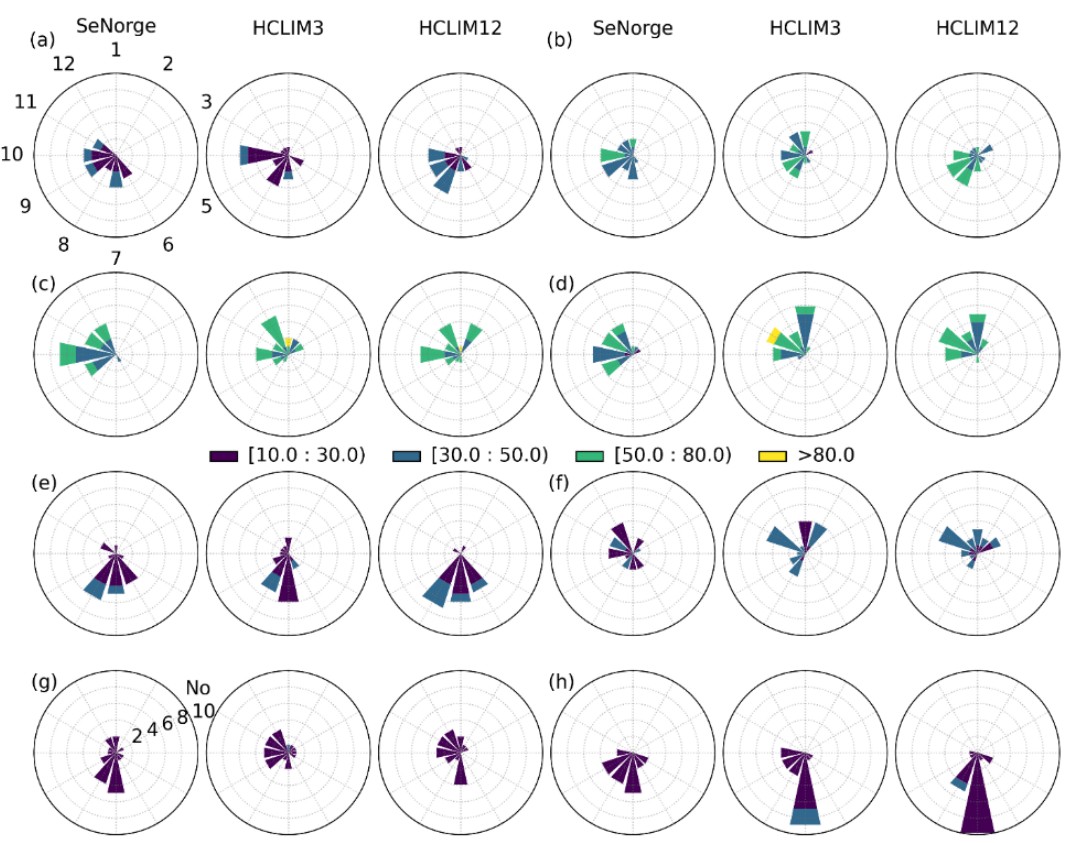


**Figure 13: The seasonality of frequency and magnitude of Rx1d precipitation from the SeNorge, HCLIM3 and HCLIM12**
**during 1999-2018 over different regions: a) Eastern, b) Southern, c) South-Western, d) Western, e) Middle-Inland, f)**
**Middle-Coastal, g) Northern-Coastal, h) Northern-Inland.**

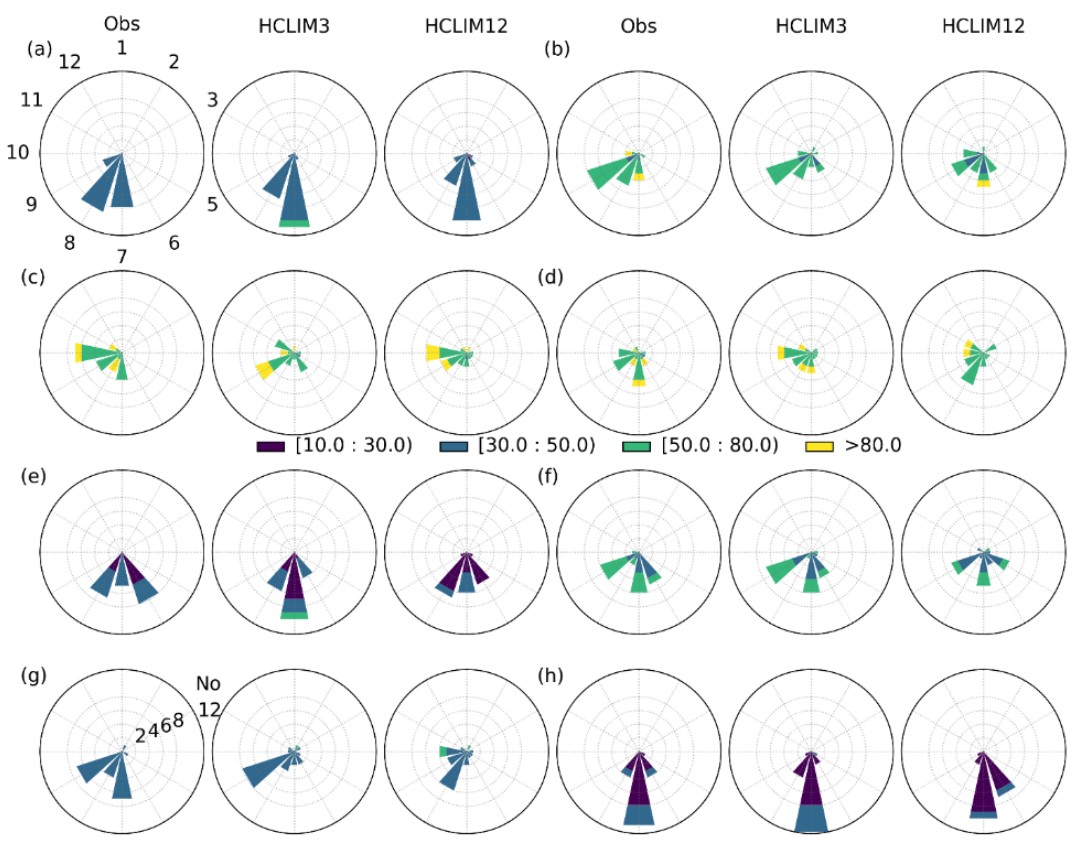

**Figure 14: The seasonality of frequency and magnitude of Rx1d precipitation from the in-situ observation, HCLIM3 and HCLIM12 during 1999-2018 over different regions: a) Eastern, b) Southern, c) South-Western, d) Western, e) Middle-Inland, f) Middle-Coastal, g) Northern-Coastal, h) Northern-Inland.**

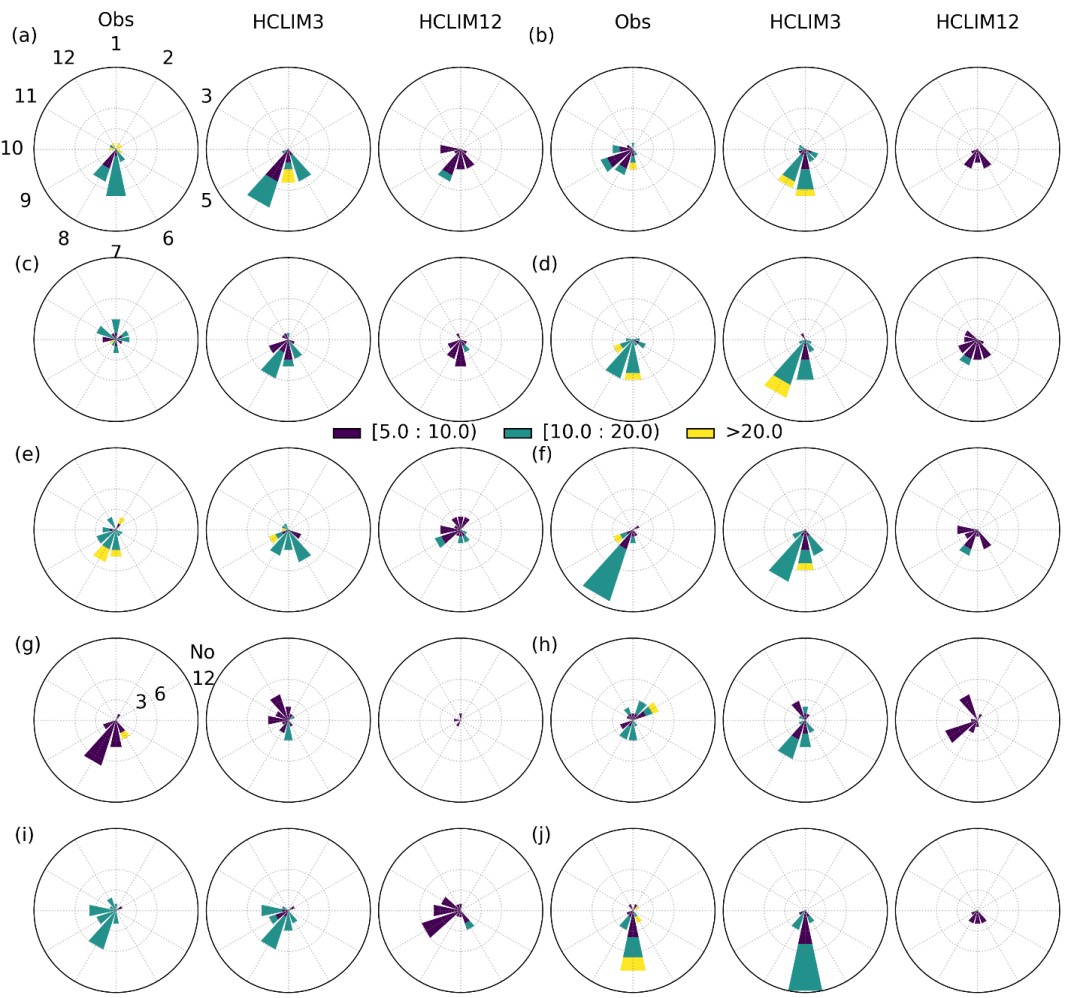

**Figure 15: Seasonality of the frequency and magnitude of Rx1h precipitation from the in-situ, HCLIM3 and HCLIM12 during 1999-2018 at 10 rain gauge stations (Table 1), i.e., a) SN11500, b) SN69150, c) SN58900, d) SN17870, e) SN44300, f) SN12550, g) SN90400, h) SN76530, i) SN56420, j) SN23500.**

Figure13 and Figure 14 show the comparison of seasonality (e.g., frequency and magnitude) of Rx1d from HCLIMs compared to SeNorge and in-situ observation, respectively. From the seasonality of daily extreme precipitation in Fig. 13, we can see that winter-autumn precipitation is dominant in almost all regions, except the Middle-Inland and Northern-Inland regions where spring-summer precipitation is dominant, similar results can be found in HCLIM3. However, the spring-summer dominant precipitation is observed in the Eastern region from HCLIM12. Heavy precipitation over 50 mm/day occurs mainly in the Southern, South-Western, and Western regions which is also simulated by both HCLIM3 and HCLIM12. In general, both HCLIM3 and HCLIM12 demonstrate competence in capturing the magnitude of extreme daily precipitation seasonally across all regions. Particularly noteworthy is the enhanced capability of HCLIM3 in capturing the seasonality of extreme precipitation frequency over the eastern,





south-western, middle-inland, middle-coastal, and northern-coastal regions compared to HCLIM12. This superior
representation of the frequency of extreme daily precipitation in HCLIM3 is consistently evident on a seasonal basis
relative to HCLIM12. The performance of the seasonality of Rx1d from HCLIM3 and HCLIM12 in the regional
scale is also confirmed by the in-situ observation (Fig. 14).

Examining the seasonality of Rx1h at local scale in Fig. 15, we can see that HCLIM3 provides a more accurate

representation of the seasonality of Rx1h in comparison to HCLIM12, which fails to produce the Rx1h. However,
comparing with in-situ observations, HCLIM3 tends to overestimate the frequency. For example, in SN58900 and
SN56420 located in the Western region, HCLIM3 simulates more frequent events, although the magnitude of
precipitation is underestimated. Both observations and HCLIM3 indicate that winter-autumn is dominant in the
Western region. The CPRCM excels in reproducing the Rx1h, surpassing the RCM in both regional and station
scales, particularly at localized scale.
**4.5 Orographic effect on seasonal extreme precipitation**
**4.5.1 Seasonal Rx1d at regional scale**

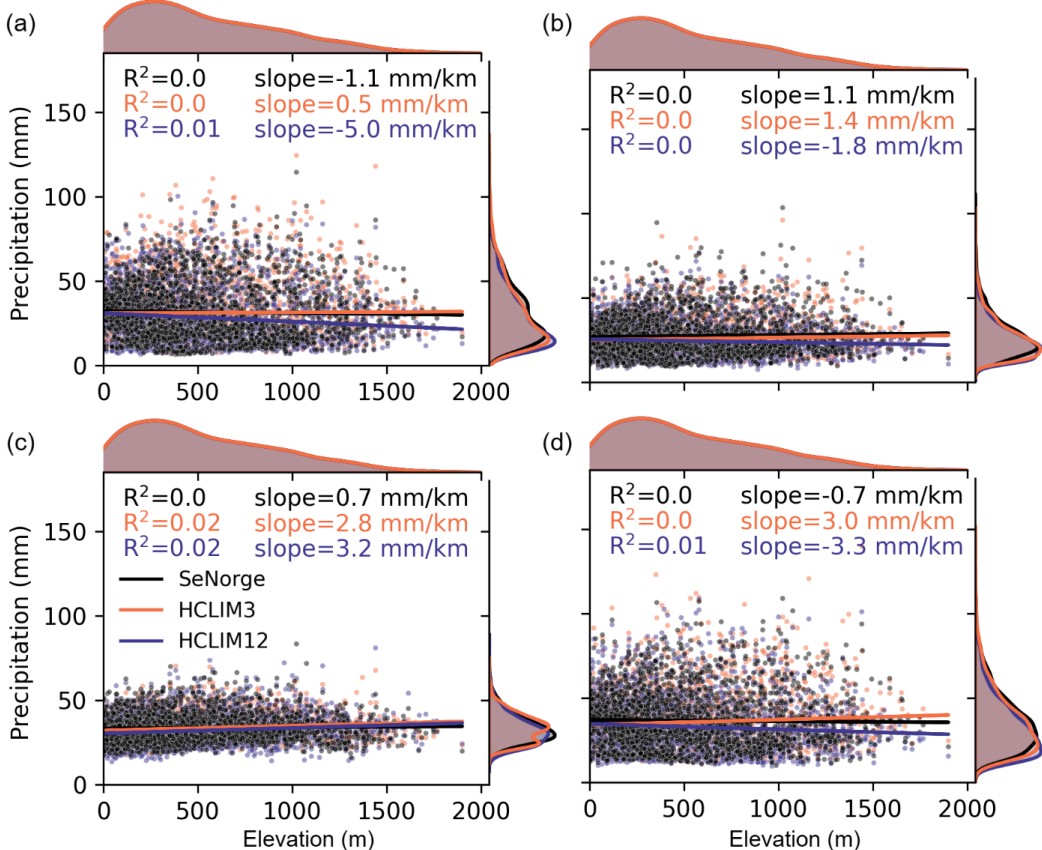

**Figure 16: Relation of (a) winter, (b) spring, (c) summer, and (d) autumn Rx1d from SeNorge and HCLIMs (i.e.,**
**HCLIM3 and HCLIM12) with elevation over Norway between during 1999-2018.**






The relationship of seasonal Rx1d with elevation from SeNorge, HCLIM3 and HCLIM12 is shown in Fig. 16. We
can see that HCLIM3 more accurately captures the no evident linear relation (indicated by zero coefficient of
determination) of seasonal Rx1d with elevation, similar to SeNorge, though it depicts a more pronounced increase
with elevation than SeNorge during summer. For example, both HCLIM3 and HCLIM12 simulate a large average
elevation-related increase in summer Rx1d (over 2.8 mm/km). Conversely, HCLIM12 reflects the reverse
orographic effect in winter and autumn Rx1d, showing a more significant decrease in Rx1d with elevation than
observation. The variation of Rx1d with elevation in HCLIM12 is consistently larger than observation, as indicated
by the larger absolute slope values.
**4.5.2 Seasonal Rx1d at local scale**

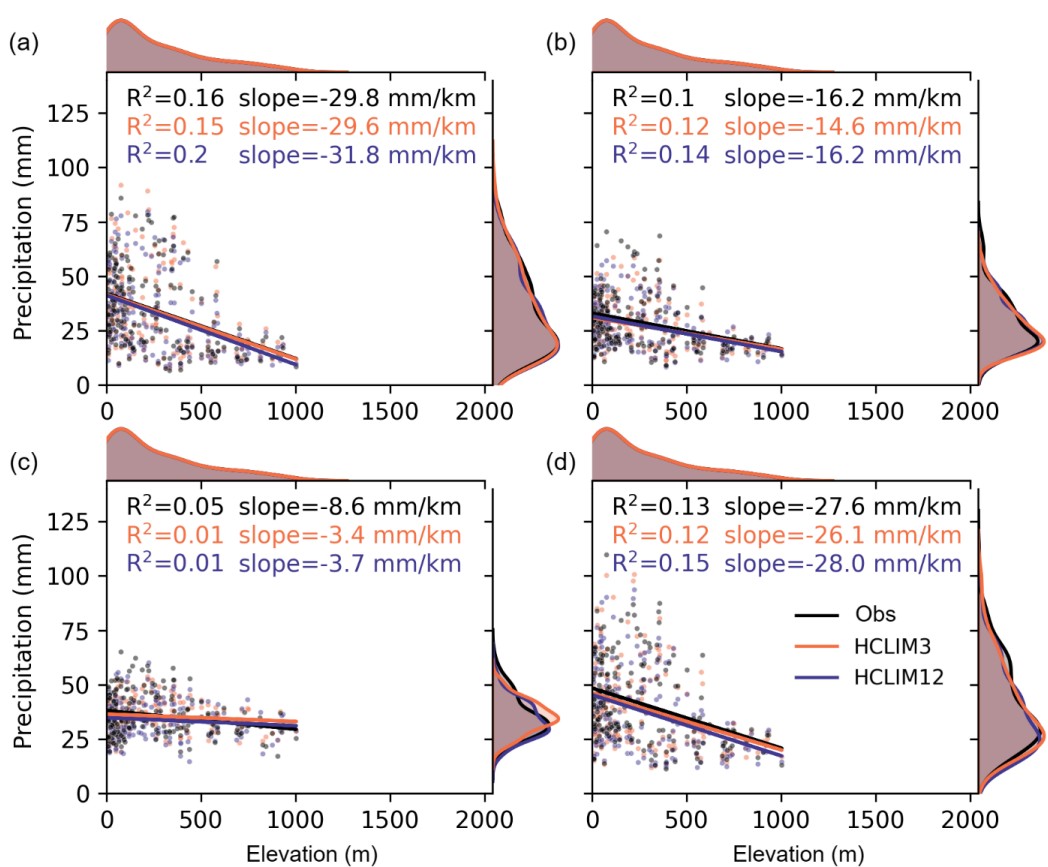

**Figure 17: Relation of (a) winter, (b) spring, (c) summer, and (d) autumn Rx1d from daily in-situ observation and**
**HCLIMs (i.e., HCLIM3 and HCLIM12) with elevation over Norway between during 1999-2018.**





Figure 17 represents the relationship of the seasonal Rx1d from in-situ observation, HCLIM3 and HCLIM12 with
elevation at local scale. The observed reverse orographic effect, seasonal Rx1d decrease with elevation, clearly
depicts with an average decrease of winter, spring, summer and autumn Rx1d of more than 29.8, 16.2, 8.6 and 27.6
mm/km. HCLIM3, although with smaller slope than observation, shows improvement in capturing the orographic
effect on winter Rx1d than HCLIM12. Moreover, HCLIM12 displays a more pronounced decline in Rx1d with
elevation, as evidenced by a steeper slope, across all seasons except summer, when compared to observation.
Despite of this, HCLIM12 more accurately represents the orographic influences on Rx1d in all seasons except
winter. Furthermore, as depicts in Fig. 17, the decreasing density of stations with increasing elevation complicates
the assessment of orography's impact, thereby challenging the reliability of our evaluations in elevated terrains.

**4.5.3 Seasonal Rx1h at regional scale**

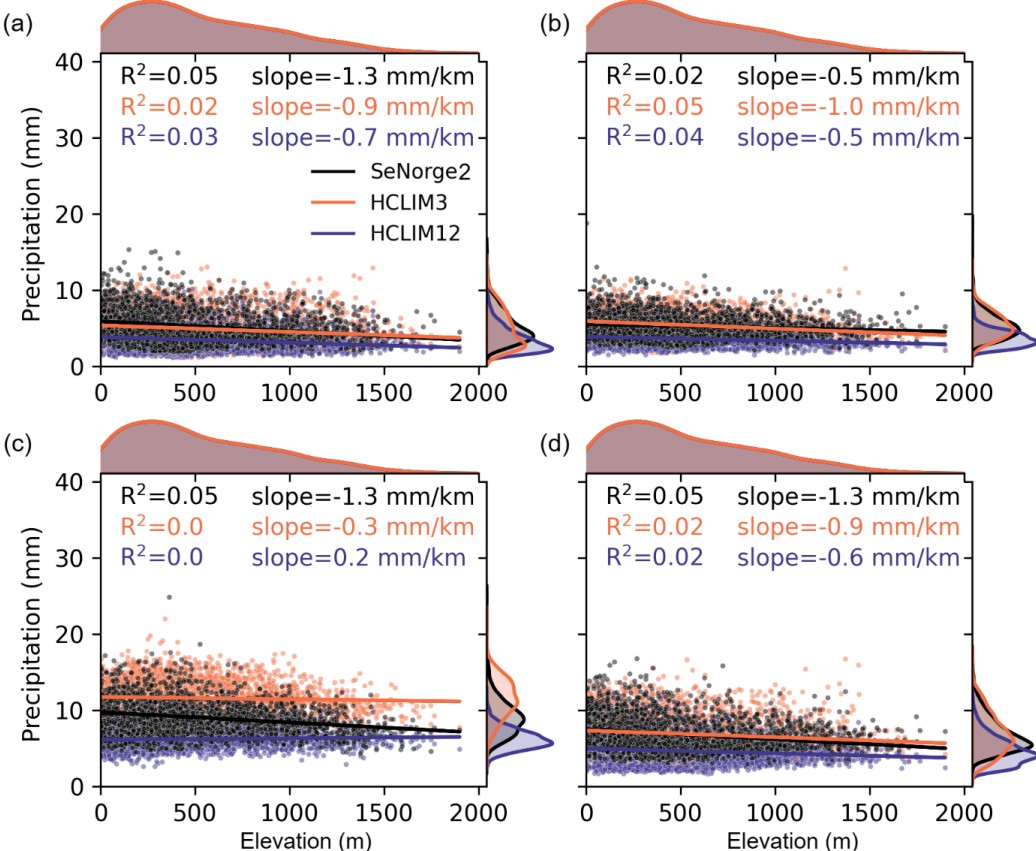


**Figure 18: Relation of (a) winter, (b) spring, (c) summer, and (d) autumn Rx1h from SeNorge2 and HCLIMs (i.e.,**
**HCLIM3 and HCLIM12) with elevation over Norway between during 2010-2018.**





The relationship of seasonal Rx1h with elevation from gridded observation and simulation is further explored, as
shown in Fig. 18. Different with seasonal Rx1d at regional scale, the reverse orography effect on Rx1h from
SeNorge2 clearly emerges in all seasons, with an average decrease of more than 1.3, 0.5, 1.3 and 1.3 mm/km in
winter, spring, summer and autumn, respectively. HCLIM3 more accurately captures the pronounced decrease in
seasonal Rx1h with elevation during winter, summer and autumn, though it still underestimates the decrease relative
to actual observation. By contrast, HCLIM12 can only reflect the similar reverse orographic on Rx1h with
observation in spring. The density plots of Rx1h reveal a significant dry bias in HCLIM12, particularly noticeable in
summer, where it inversely correlates Rx1h with elevation. HCLIM3 more effectively represents the reverse
orographic impact on hourly precipitation than on daily, as seen by comparing the slope and precipitation
distribution in the Fig. 16 and Fig. 18.

**5 Discussion**
**5.1 Added value of CPRCM at regional scale**
The comparison between HCLIM3 and HCLIM12 reveals distinct biases in the representation of Rx1d, particularly
marked by an orographic effect. HCLIM3, on average, exhibits lower bias compared to HCLIM12, with notable
overestimation of precipitation along coastal areas. Converse to the wet-bias from HCLIM3, HCLIM12 shows
underestimation in the complex orography for the annual Rx1d. This discrepancy is attributed to unrealistic cloud
concentration nuclei numbers, while complex terrain areas also experience overestimation of precipitation from
HCLIM3 due to micro-physics, confirming findings from Lind et al. (2020). The comparison in Fig. S2 for annual
precipitation also represents large difference, particularly marked by a coastal-mountain-inland division. HCLIM3,
on average, exhibits a slightly larger wet-bias compared to HCLIM12 over the complex terrain, which could be
related to the model physics confirmed by Lind et al. (2020).
Notably, CPRCM (i.e., HCLIM3) demonstrates more significant benefits in capturing Rx1d at numerous grids
in Norway compared to SeNorge. Spatial variations in precipitation patterns across eight regions in Norway further
highlight the nuanced performance of HCLIM3, particularly in complex topography areas. Consistent with the
observation from Dyrrdal et al. (2015), HCLIM3 also displays a lower bias than HCLIM12 on average, aligning
with the observed strong west-east gradient of precipitation over complex terrain. Dyrrdal et al. (2015) and Poujol et
al. (2021) emphasize the diverse mechanisms driving precipitation in different regions of Norway. For instance,
extreme precipitation in the western region is dominated by frontal systems and orography during autumn and
winter, while convective activity plays a crucial role in the southern regions during summer (Li et al., 2020b).
HCLIM3, however, slightly overestimates annual Rx1d in complex terrain, indicating challenges in convection
parameterization schemes.
The study by Thomassen et al. (2023) utilizing SPHEAR with a 2.2 km resolution echoes challenges in
accurately simulating extreme precipitation over complex terrain in the northern Italy. Our findings confirm the





benefit of convection-permitting models in capturing spatial distribution in complex terrain, even though slight
positive biases persist when compared with SeNorge data.
A potential reason for the observed biases is related to the limitations of pseudo-observation SeNorge, which
may inadequately represent steep valleys at the 3 km grid space (Thomassen et al., 2023). Sparse station distribution
in complex mountain areas may contribute to unrealistic observations, concealing the benefits from HCLIM3. The
future work should delve into biases introduced by SeNorge and its potential impact on observed precipitation
patterns. Comparisons between SeNorge and HCLIM3 in data-sparse areas should be approached with caution due
to uncertainties associated with the SeNorge dataset.
The comparison of HCLIM3 and HCLIM12 in terms of seasonal Rx1d reveals better representation by
HCLIM3, except for a dry bias in the southwest region. This dry bias could be attributed to the limitations of
HCLIM3 in capturing specific precipitation mechanisms in this region.
Furthermore, HCLIM3 shows obvious benefit in capturing Rx1h on average than HCLIM12, even with wet-
bias from HCLIM3 compared to hourly SeNorge2 over Norway. Most grids from HCLIM12 underestimate the
Rx1h, indicating the misrepresentation from parameterization schemes, confirming the finding from Lind et al.
(2020), more "drizzle" in HCLIM12. The uniformed spatial variation for the annual Rx1h may be attributed to the
too sparse station distribution (Lind et al., 2020), resulting in the damping for local storms. The station density
induced error from gridded dataset has also been indicated in Gervais et al. (2014b), who suggested the source for
large errors in gridded dataset when station density is low.
In summary, HCLIM3 demonstrates better agreement with observations at regional scales in Norway
compared to HCLIM12. This is consistent with previous studies highlighting the superiority of convection-
permitting models, especially in capturing extreme precipitation events over complex terrain.

**5.2 Added value of CPRCM at local scale**
The examination of CPRCM at the local scale reveals its superiority representation of extreme precipitation. Our
analysis focuses on in-situ observations (at local scale), recognizing the limitations of aggregation, regional average
or pooling techniques at grid scale, as highlighted by Kendon et al. (2023).
Chapman et al. (2023) compared return-levels between an Africa convection-permitting climate model with
4.5 km grid-spacing (CP4A) and observation at station-level and grid and found large differences. CP4A closely
aligns with station return-levels but exhibits a slight undercatch for the precipitation. Our results at station scale
corroborates these results, as return levels consistently underestimate extreme precipitation in HCLIM3. In line with
findings in Malawi, where regional-climate-model (P25) overestimated return levels with increasing return periods.
The damped extremes from averaged precipitation within the gridbox may cause smaller return-level at grid scale.
Therefore, significant bias of HCLIM3 and HCLIM12 relative to observation between local scale (station-level) and
regional scale may also be attributed to the damped extremes. The hourly and daily precipitation extremes from
HCLIM3 at most stations show more realistic results than at the regional scale, supporting the hypothesis of damped
extremes at regional scale weakening the superior, even though more bias is observed than regional result.


Despite the higher uncertainty associated with extremes from CPRCM at local scale (Chan et al., 2020;
Cannon et al., 2019), our results demonstrate smaller biases in HCLIM3. The higher extreme damping in HCLIM12,
averaging over coarse resolutions (~12km), may contribute to its higher dry bias. Furthermore, the credibility of
CPRCM in providing reliable representation for extremes may attribute to resolve deeply convective activities at
sub-grid scale, while the deficiency in the convection-parameterization scheme (HCLIM12) may bring uncertainty.
In contrast to the numerical instability in RCMs noted by Kendon et al. (2023), we note underestimation of
extremes, including return-level and time evolution in HCLIM12 at local scale, attributable to its convection-
parameterization scheme. In summary, the added value of HCLIM3 in capturing the frequency of extreme
precipitation at station scales, especially at highly localized local scale, is evident when compared to HCLIM12.
Few studies have compared sub-daily and daily rainfall in RCMs due to its unreliable prediction at sub-daily
scale. Following Ban et al. (2014), we recognize that the added value of CPRCM in capturing sub-daily or daily
precipitation extremes is different. The performance of RCM ~10 km in representing sub-daily rainfall was limited,
which has been proved hard to capturing sub-daily extreme rainfall in the southwestern United States (Jiang et al.,

2013).

Contrary to sub-daily extremes, HCLIM12 demonstrates higher merit in capturing daily extreme precipitation,
with biases less than 50%. By comparing the Rx1d/Rx1h at local scale, we also note that the added value of
HCLIM3 in capturing the Rx1h is more obvious. The superior performance at hourly scale is consistent with
findings by Médus et al. (2022) and Ban et al. (2014), emphasizing the significantly better sub-daily precipitation
characteristics of CPRCM, including spatial distribution and duration-intensity features.

### 607 5.3 Seasonality of extreme precipitation

The performance of CPRCM in capturing the seasonality of precipitation driven by different physical process has
been a subject of investigation in previous studies. Moustakis et al. (2021) highlighted the adequacy of CPRCM
(CTL-WRF~4 km) in capturing observed seasonality in the United States. Prein et al. (2013) also emphasized
CPRCM's ability to better reflect summer precipitation due to stronger deep convective activity. They found there
was less difference between RCM and CPRCM in capturing winter precipitation.
In our comparison between HCLIM12 and HCLIM3, we observe superior representation of hourly
precipitation seasonality in HCLIM3. The frequency and intensity of Rx1d in regional scale and Rx1h at station-
level can be better captured by HCLIM3, especially for local scale. The varying response to seasonal extreme
precipitation across different regions with distinct climate characteristics suggests that CPRCM tends to perform
better in specific regions or seasons with more convective precipitation. This aligns with finding from Prein et al.
(2013), emphasizing CPRCM's strength in capturing convective processes.
Further investigation of CPRCM and RCM performance in different regions regarding extreme precipitation
reveals the added value of HCLIM3, especially when assessing precipitation frequency and intensity on a seasonal
basis. This advantage is more pronounced at the local scale compared to the regional scale. However, investigating
daily precipitation intensify and frequency at local scale reveals unexpected results, notably the intensify of Rx1d
during winter is better captured, while the southern and south-western regions with more convective activity during





summer are poor represented by HCLIM3. This suggests that HCLIM3's performance is not solely constrained by
specified convective precipitation, and the consistent underestimation of Rx1d intensity during summer in
HCLIM12 may be attributed to higher uncertainty from its convective parameterization scheme or numerical
uncertainties at the local scale.

**5.4 Added value of CPRCM in reproducing orographic effect**

At the local scale, seasonal Rx1d decrease with elevation, known as reverse orographic effect, is captured by
HCLIMs in all seasons. Compared to HCLIM12, HCLIM3 shows added value in representation of reverse
orographic effect during winter. In other seasons, however, HCLIM12 shows more improvement in the relation of
extreme precipitation with elevation than HCLIM3. It is noteworthy that the differences in how seasonal Rx1d
relates to elevation are more pronounced at a regional scale than at a local scale.

At regional scale, no evident relation of Rx1d with elevation is found from observation in all seasons, which

have also been shown from HCLIMs except summer. Furthermore, the unclearly relation of seasonal Rx1d with
elevation at regional scale was also seen from the study of Dallan et al. (2023), in which, they analyzed annual Rx1d
based on CPRCMs and in-situ observation over Alpine. In summer, HCLIMs shows the orographic effect on Rx1d,
which may be explained by the overestimated orographic precipitation from HCLIMs. Different to the unclearly
orographic impact on Rx1d at regional scale, steeper slope is clearly seen than that from the local scale. Conversely,
the so-called "orographic enhancement" (Avanzi et al., 2021), precipitation increase on the windward side induced
by the lifting of air masses, and decrease along the leeward side due to air descent and drying is not found here at the
regional or local scales.

For hourly extremes, the results of reverse orography effect on seasonal Rx1h are consistent with previous

study of Dallan et al. (2023), which also found the weak decrease of annual Rx1d from CPRCMs to elevation than
in-situ observation over Alpine. HCLIM3 and HCLIM12 well capture the reverse orography effect on Rx1h,
especially in HCLIM3, although a stronger decrease of Rx1h with elevation is observed from SeNorge2 except
spring. Besides, lower Rx1h and weak reverse orography effect is found in HCLIM12 in all seasons. The orographic
effect on hourly and daily extremes seasonally suggests the influence of orography on extreme precipitation at
different timescales, and highlights the reliable simulation of extreme precipitation over complex orography. Our
findings confirm the reverse orographic effect on Rx1h, as previously observed for hourly precipitation (Marra et al.,

2021).

In summer, the poor performance from HCLIM3 and HCLIM12 in capturing orography effect and extreme

precipitation may be related to the intense orographically-sustained convection affected by atmospheric, aerosol
conditions, local terrain slope and shadowing effects, which failed to be captured by 3 km CPRCMs (Dallan et al.,
2023; Poujol et al., 2021). Moreover, Marra et al. (2021) also confirmed that the reverse orography effect on short-
duration precipitation extremes could be attributed to a weaking of updrafts of moist air over mountain ridge by
orographic turbulence. However, another major source resulting in the bias between HCLIMs and observation could
be related to the observation uncertainty. Sparseness of hourly stations and undercatch problems could also lead to





underestimation and underestimation of precipitation, especially in the complex orography (Lussana et al., 2018,
2019).

Furthermore, we acknowledged that the linear regression method is closely related to the distribution of data
point. More specifically, the different relationship between seasonal Rx1d and elevation may be attributed to the
uncertainty from the observed data. For example, there is limited rain gauges over the complex orography, the
highest rain-gauges are located about 1000 m. Importantly, several land surface characteristics could influence the
precipitation, a multiple linear regression model should be considered to quantify the orographic effect more realism
(Zhang et al., 2018). Besides, there might be uncertainty from SeNorge gridded dataset. Therefore, the relationship
between Rx1d and elevation remains inconclusive. Besides, the reverse orographic effect from Rx1h also needs to
be improved due to the lack of sufficiently long data records. Enhanced observation and more comprehensive
datasets are necessary to solidify our understanding of this connection.

## 671  6 Conclusions

In this study, we conducted a comprehensive evaluation of extreme precipitation characteristics from regional to
local scale in Norway, focusing on eight distinct regions, utilizing a convection-permitting regional climate model
(HCLIM3) and comparing it with its convection-parameterized regional climate model (HCLIM12) forced by ERA-
Interim data during 1999-2018.
The key conclusions drawn from this study are as follows:
a) For daily extreme precipitation at regional scale, HCLIM3 shows benefit in capturing Rx1d compared to
HCLIM12, showcasing improved representation of Rx1d in all seasons in the most of regions except
south-western. The added value of HCLIM3 varies across regions, demonstrating its superiority in
capturing return levels and daily extreme precipitation frequency, particularly in the southern, middle-
inland, middle-coastal, and northern-inland regions based on both SeNorge and in-situ data. HCLIM3
shows greater potential in reproducing the frequency of daily extreme precipitation exceeding 10, 15, and
20 mm.
b) For hourly extreme precipitation at regional scale, HCLIM3 also shows superiority on average, with wet-
bias based on SeNorge2, compared to HCLIM12. Furthermore, the added value from HCLIM3 in
capturing seasonal Rx1h is also observed than HCLIM12 in all seasons and regions except western,
middle-inland and middle-coastal regions during summer. In contrast, the overestimation of Rx1h from
HCLIM12 across eight regions in Norway is found.
c) For daily extreme precipitation at local scale, the Rx1d feature including frequency, intensity and return
level in most regions can be better captured by HCLIM3 than HCLIM12, although the benefit from
HCLIM3 over HCLIM12 diminishes in western region at local scale compared to that at regional scale.
Except south-western, HCLIM3 also have not superiority in capturing Rx1d in the western region.
d) For hourly extreme precipitation at the local scale, HCLIM3 outperforms HCLIM12 in capturing the
annual variability of Rx1h during 1999-2018, although the shifting of peak occurrence or magnitude is



observed than observation at some stations. Compared to HCLIM12, the add-value of HCLIM3 in
capturing Rx1h is more obvious at local scale than regional scale. The extreme precipitation characteristics
including frequency, intensity and return-level from HCLIM2 at local scale are underestimated seriously.
Besides, HCLIM3 also shows more added value in capturing Rx1h than Rx1d from regional to local scale.
e)    For the seasonality of extremes, our analysis reveals no substantial difference between HCLIM3 and
HCLIM12 when it comes to daily extremes. However, a distinct advantage emerges with HCLIM3 for
hourly extremes, where it accurately reflects both the occurrence and intensity of these events across
different seasons. On the other hand, HCLIM12 tends to underestimate these aspects, demonstrating a
significant bias in capturing the frequency and magnitude of hourly extreme
f)    The reserve orographic effect on seasonal Rx1h at regional scale emerge in Norway and can be better
captured by HCLIM3 than HCLIM12 except spring, although a stronger decrease is found in observation.
Additionally, a significant reverse orographic effect on seasonal Rx1d at the local scale has been observed,
with HCLIM3 providing added value, especially in capturing winter Rx1d. However, the relationship
between seasonal Rx1d and elevation at the regional scale remains ambiguous.

## Acknowledgments

We would like to thank Stefan P. Sobolowski and Ozan Mert Gokturk, for their great supports as PI and data
manager of the EU Impetus4change (I4C) project, respectively. This research was supported by the European
Union's Horizon 2020 research, innovation programme under grant agreement no. 101081555
(IMPETUS4CHANGE) and the Research Council of Norway through FRINATEK Project 274310. The computer
resources where available through the RCN's program for supercomputing (NOTUR/NORSTORE); projects
NN10014K and NS10014K. All simulation data in this paper are available from the authors upon request
(luli@norceresearch.no).

*Competing interests.* The authors declare that they have no conflict of interest.

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
