# Peer review of "Enhanced Evaluation of Sub-daily and Daily Extreme"

_Hydrology and Earth System Sciences, 2024_

## Author Comment (AC1)

**REPLY TO THE COMMENTS OF THE REFEREE #1**

**Dear reviewer,**

**First of all, we would like to thank you for the time you have spent reviewing our manuscript. We strongly appreciate the constructive comments and valuable feedback made. We have carefully addressed the reviewer's comments and suggestions. Below are our point-by-point responses to the comments in blue.**

**Thank you very much again for your review.**

**Author and Co-Authors**

**COMMENTS FROM REVIEWER#1**

The paper presents an evaluation of extreme precipitation from two regional climate models; one is a regional model at 12km resolution, the other is a convection-permitting model at 3km resolution. The evaluation is based on different observational products: two grid products at 1day and 1hour temporal resolution, and a network of rain gauges (about 190 at daily resolution and 10 at hourly resolution). Seasonality and relation with elevation are also explored, at regional (by contrast with the gridded product) and local scale (by contrast with rain gages). The main findings indicate generally better performance for the HCLIM3 than the HCLIM12 model, more clearly at hourly resolution.

The study is of interest on the general topic of evaluation of extreme precipitation from convection-permitting models, which may help in better understanding how to use them in practical applications. In my opinion, it gives an incremental advancement in this field more than novelty, considering the regional scale (Norway) and the use of metrics commonly used in these kind of studies (Rx1d, Rx1h, return levels). It is well written, but I found difficult to get the main messages on the results because of the total length, number of figures and panels. The topic is of interest, and fitting the journal scopes, but I suggest a few major revisions and some minor before publication in HESS. My comments are listed below.

- Major 1. You used 8 regions. I wonder if less can be used, pooling together smaller ones. I say this based on two considerations: 1) extension is very different across the regions, and regional evaluation of models are then based on quite different number of grid points (for example, how many grid points for the two small regions in the south?) and number of daily rain gages (just 4-11-14 for three regions!); 2) it is difficult to follow the explanations and figures with comparisons on 8 regions, 2 models, 4 seasons, 2 durations ... and get a message on the results; maybe having less could help.

Reply: Thanks for your valuable comment. Based on your suggestions, we conducted additional analyses by merging the two smaller regions in the south (southern and south-western) and the two regions with fewer rain gauges (northern-coastal and northerninland). We have updated Figure1, 2 and 4 accordingly (please see the revised figures below). Our finding from these new analyses indicated that:

(1) HCLIM3 performed worse than HCLIM12 in seasonal and annual daily maximums (Rx1d) in the newly merged regions. This contrasts with our original findings using eight regions, where the deficiencies of HCLIM3 were primarily observed in the south-western region alone at regional scale. The discrepancies are noticeable in the annual and seasonal Rx1d biases (Figures R1 and R2) and in the biases of extreme annual Rx1d at various return levels (Figure R3). For example, the positive bias in the south-western region is compensated by the negative bias in the southern region in the merge region of S-SW (southern and south-western) during winter, leading to a different interpretation of HCLIM3's added value compared to HCLIM12. Overall, HCLIM3 does not show added value in the south-western region seasonally and merging it with southern region masks the potential benefits of HCLIM3 observed in the southern region alone.

(2) Hanssen-Bauer et al. (2006) originally divided Norway into 13 regions, which were further reduced to 8 regions based on shared characteristics, as suggested by Konstali and Sorteberg (2022) and Michel et al. (2021). Given Norway's complex coastal climate and geological conditions, 8 regions represent the minimum number necessary to capture the climatic variability adequately.

(3) The overall goal of the supported Impetus4Change (I4C) project is to improve the usability of climate information and services at the local and regional levels. To provide

climate information for each administrative region in Norway directly, we divide Norway into 8 regions.

Therefore, we decided to retain the original eight regions in our study, as recommended by Konstali and Sorteberg (2022) and Michel et al. (2021).

We acknowledge that the current presentation of results may be overwhelming and require better synthesis to clarify conclusions and convey a clear message. To address this, we have improved the presentation of our findings in the revised manuscript. For example, we identified redundant in Figure 5, 9, and 12 compared to Figure 13, 14, and 15, which already include precipitation frequency data. Consequently, we have removed Figure 5, 9, and 12 to reduce redundancy and enhance clarity. This streamlining aims to make our conclusions more concise and easier to follow.

[Figure]

**Figure R1: The percentage bias of seasonal Rx1d from HCLIM3 and HCLIM12 to SeNorge for merged southern and south-western (S-SW), northern-coastal and northern-inland (NCo-NI) regions.**

[Figure]

**Figure R2: The percentage bias of seasonal Rx1d from HCLIM3 and HCLIM12 to daily in-situ observation for merged southern and south-western (S-SW), northern-coastal and northern-inland (NCo-NI) regions.**

[Figure]

**Figure R3: The bias of extreme annual Rx1d exceeding the 5-year to 50-year over eight regions between seNorge and HCLIMs (i.e., HCLIM3 and HCLIM12).**

- Major 2. Regional scale is here referred to the analysis using the gridded products; local scale is referred to the analysis based on rain gages. They show different results, but I wonder how much this is due to the use of different observation products. How is SeNorge-models comparison sampled on the same rain gage points? Or, how is seNorge compared to rain gages? Differences you highlight in your text (e.g lines 337, 352, 384, etc) could be due to the use of a different observational benchmark. Moreover, when comparing gridded products and

climate model, you compare values from a same size grid (12x12km), when

comparing model and rain gages, the comparison is made on a 12x12km grid and

a point measurement. I suggest to add further analysis comparing seNorge with

rain gages, or extracting the comparison between seNorge vs models on the

same locations of rain gages, or to add some considerations in the discussion.

Reply: We fully agree with your comment. We have analyzed and compared seNorge with

rain gages by interpolating the seNorge to the same location of rain gauges. We have

added it in the discussion in the revised manuscript.

- Major 3. Biases (e.g. figure 2, 6) are shown in mm (absolute differences). Maybe

  relative differences (modelled-observed)/observed could be more meaningful

  considering that precipitation has a big range across the regions: a bias of 5 mm

  is different on a 30 mm or a 80 mm daily precipitation! My suggestion is to update

  maps and plots of biases with relative bias (%), and to revise description of results

  and comments on "magnitude" of bias based on this.

Reply: Thanks for your comment. We have updated all the plots with relative bias (%).

Minor comments

1. title: I suggest hourly in place of sub-daily, considering that you just evaluate 1h

   Reply: Thanks for the comment, the titles were changed for: Enhanced Evaluation of hourly and Daily Extreme Precipitation in Norway from Convection-Permitting Models at Regional and Local Scales.

2. Line 120: these results on orographic effect "were based on the annual maxima"... yes. But also yours are based on Rx1d (see line 222-223). So ... why do you highlight these about other studies at line 120, if then you do the same? I suggest to remove.

   Reply: Thanks for your comment. We have removed it in the revised manuscript.

3. Line 121-122. "The dependence on seasonality ... need the evaluation based on season". Of course! Maybe you wanted to tell something different here and I didn't get it. Please calrify/modify.

   Reply: Thanks for your comment. We appreciate your feedback. What we intended to convey is that the performance of CPRCMs is heavily influenced by seasonal variations, which necessitates evaluating the orographic effects of seasonal extremes in addition to annual extremes. This is crucial for understanding how different seasons impact model performance and the resulting hydrological responses.

Line 195. Not clear: you say here you averaged the indices in the region ... then in caption of figure 2 you write the bias is calculated at each grid point (this makes sense to me). So, when do you use the averaged indices?

Reply: Thanks for your comment. To clarify, the bias is initially calculated at each grid point within the region. After calculating the bias for each grid point, we then average these biases across all grid points within each region to obtain a regional average bias. This approach ensures that the regional bias is representative of all grid points in the region.

4. Line 206. Specify somewhere in the section that this is done for daily data (both gridded and rain gages), while just on 10 rain gages at 1h duration. More importantly, later in the text (lines 367, 372,..) you mention uncertainty ... and it is never explained before in the paper how it is evaluated. Add it in the methodology.

Reply: Thanks for your comment. The uncertainty means the long whisky. We have rewritten it more clearly in the revised manuscript.

5. Figure 2 (same for figures 6 and 7). The caption mentions absolute bias but describes a relative bias calculation. Please correct.

Reply: Thanks for your comments. We have corrected it to percentage bias (simulations minus observations, divided by observations) in the revised manuscript.

6.  Figure 2c (same for figures 6 and 7). I suggest to add another row with the regional bias for annual Rx1d, not just seasonal, in order to have a synthesis of what is shown in the maps in panel a.

    Reply: We fully agree with your comment. We have added the row with regional bias for annual Rx1d in the revised manuscript.

7.  Figure 4. Same y-axis limits could help in comparing the bias... and maybe you can revise the description of results better considering the different magnitude of the bias (example at lines 290-291).

    Reply: We agree. We have updated the plot with same y-axis limits.

8.  Figure 4. I can't understand why Northern-Coastal has so big bias for HCLIM3, considering that figure 3 shows a tendency of underestimation of the empirical distribution, similar to Northern-Inland, for which you find big underestimation of return levels. And for the Norther-Inland, why underestimation of return levels is bigger for HCLIM3, while the distribution in figure 3 show more underestimation for HCLIM12? ... please check the correctness of the results.

    Reply: Thanks for your comment and for pointing out the discrepancies. Upon review, we recalculated the return levels for all regions and found that the calculations for the Northern-Coastal and Northern-Inland regions were indeed incorrect. As a result, we have corrected these errors and updated the results and corresponding plot. We apologize for the confusion. Thank you again for bringing this to our attention.

9. Line 370. For 50yr return-periods they seem identical, not larger bias for HCLIM3. I would remove it

   Reply: Thanks for your comment. The 50yr return-periods has now removed.

10. Line 439-440. This is based on just 10 points. This can't be considered a general finding, I suggest to mention the limit of the analysis.

    Reply: We fully agree with your comment. We have added the analysis of the limitation in the corresponding results and discussions.

11. Line 483, section 4.5. You show the slope of precipitation with elevation as absolute value, mm/km. I strongly suggest to calculate and show it as relative slope, for example with respect to the average value of Rx. Because 1mm/km has a different magnitude for Rx1d and Rx1h. Then I suggest to revise your discussion considering this … (e.g. I see very weak relation of Rx1d with elevation, so I'm sure you can really speak about reverse orographic effect …also at line 629)

    Reply: We fully agree. We have recalculated the relative slope and updated it in the revised manuscript.

12. Line 494. "Significant"? Based on a specific test? Maybe "relevant"…

    Reply: Thanks. We have replaced the significant with relevant.

13. Line 561-562. Not very informative consideration …. Could you elaborate more on this? Or delete …

Reply: Thanks for your comment. We have deleted it.

14. Line 580-582. I see here two contrasting points. 1) You mention *underestimation* for return levels, but for Rx1d in figure 8 I see bias around zero, while for Rx1h you have evaluation on just 10points. 2) Then you say this is in line with results in Malawi (!!!!) finding *overestimation*. I can't really understand your reasoning here.

Reply: Thanks for your comment. We apologize for any confusion caused by the presentation of our results. To clarify the two points you raised:

(1) We recognize that our initial explanation might have been unclear. For the analysis, we retained only those stations with less than 10% missing data from 1999 to 2018, resulting in a total of 10 hourly stations across Norway. Although the number of rain-gauge is small, the data quality is high and the time series is extensive. We understand that using only 10 stations introduces some uncertainty, and we acknowledge this in our revised manuscript. We have expanded our discussion to address the limitations and potential impacts of this limited dataset on our findings.

Additionally, the SeNorge2 gridded hourly data covers only an 8-year period, making it difficult to compare directly with the station-based results. However, the evaluation results derived from the 20-year station data are consistent with those from the 8-year gridded data, both indicating a significant underestimation by HCLIM12. This consistency between the short-term gridded data over the region and the long-term

station data highlights the robustness of our findings regarding HCLIM12's performance.

(2) We realize that referencing Malawi, a region with different climatic conditions, was inappropriate for drawing direct comparisons. In the revised manuscript, we have instead cited results from Thomassen et al. (2023), who studied similar return levels using the same HCLIM3 and HCLIM12 models in Denmark, which is geographically closer to Norway and more relevant in terms of climate. This citation provides a more appropriate comparison, highlighting that our findings are consistent with those observed in a comparable northern European context.

"Thomassen et al. (2023) compared the performance of HCLIM3 and HCLIM12 based on local rain-gauge data in Denmark, and found that HCLIM12 indeed underestimate the hourly extreme event and HCLIM3 agree well with observation. Our results at local scale corroborates these underestimation results limited at some local place (SN58900, SN44300, SN76539) regarding to the hourly extremes (Fig. 11), and the bias from HCLIM3 is closer to 0 in other 7 local places. Furthermore, the HCLIM3 in capturing the daily extremes (Fig. 8) at local scale tend to almost 0 bias. In addition, the return levels from HCLIM12 underestimate hourly extreme precipitation in all local places (Fig. 11) and daily extremes in southern, middle-inland, middle-coastal and northern-inland regions (Fig. 8). It should be noted that the limitation of Rx1h at 10 points, however, the obvious underestimation of RCMs in simulating the return level of Rx1h in Norway have also been indicated in Médus et al. (2022)."

15. Line 587. I can't understand the meaning of "weakening the superior" …

    Reply: Thanks for your comments. We have rewritten this sentence:

    The hourly and daily precipitation extremes from HCLIM3 at most stations show more realistic results than at the regional scale especially for hourly extremes at 10 points, supporting the hypothesis of damped extremes at regional scale uncovering the superiority from CPRCMs at some regions (eastern and middle-coastal). However, the added value from CPRCMs may also disappear for some regions (western and middle-coastal regions) at local scale compared to regional scale.

16. Line 635. Dallan et al. 2023 analyzed annual Rx1d: this can't be related with the seasonal Rx1d. I suggest to rephrase in some way: "An unclear relation of Rx1d with elevation at regional scale was also seen from the study of Dallan et al. (2023), in which, they analyzed annual Rx1d based on CPRCMs and in-situ observation over Alpine"

    Reply: Thanks for your comments. We have rewritten it according to your suggestions.

17. Line 640. I suggest to add a few recent references on orographic enhancement at daily scale observed in different regions (e.g. Formetta et al.  2021 https://doi.org/10.1016/j.advwatres.2021.104085 and Amponsah et al. 2022 https://doi.org/10.1016/j.jhydrol.2022.128090); same at line 651 for the reverse orographic effect, adding also Formetta et al 2021, considering they explored durations from subhourly to daily.

Reply: Thanks for your suggestions. We have added your suggested references in the revised manuscript.

18. Please also revise your conclusions accordingly to the modifications you will do in the revised version of the manuscript

Reply: Thank you for your feedback. We will carefully revise our conclusions to reflect the changes and updates made to the manuscript.

Thanks very much for your input, which helps us improve the quality and clarity of our manuscript.

**References**

Hanssen-Bauer, I., Tveito, O. E., & Szewczyk-Bartnicka, H. (2006). Comparison of grid-based and station-based regional temperature and precipitation series (Tech. Rep. No. 04). Norwegian Meteorological Institute.

Konstali, K. and Sorteberg, A.: Why has Precipitation Increased in the Last 120 Years in Norway?, Journal of Geophysical Research: Atmospheres, 127, 10.1029/2021jd036234, 2022.

Médus, E., Thomassen, E. D., Belušić, D., Lind, P., Berg, P., Christensen, J. H., Christensen, O. B., Dobler, A., Kjellström, E., Olsson, J., and Yang, W.: Characteristics of precipitation extremes over the Nordic region: added value of convection-permitting modeling, Nat. Hazards Earth Syst. Sci., 22, 693-711, 10.5194/nhess-22-693-2022, 2022.

Michel, C., Sorteberg, A., Eckhardt, S., Weijenborg, C., Stohl, A., and Cassiani, M.: Characterization of the atmospheric environment during extreme precipitation events associated with atmospheric rivers in Norway - Seasonal and regional aspects, Weather and Climate Extremes, 34, 100370, 10.1016/j.wace.2021.100370, 2021.

Thomassen, E. D., Arnbjerg-Nielsen, K., Sørup, H. J. D., Langen, P. L., Olsson, J., Pedersen, R. A., and Christensen, O. B.: Spatial and temporal characteristics of extreme rainfall: Added benefits with sub-kilometre-resolution climate model simulations?, Quarterly Journal of the Royal Meteorological Society, 149, 1913-1931, https://doi.org/10.1002/qj.4488, 2023.

---

## Author Comment (AC2)

REPLY TO THE COMMENTS OF THE REFEREE #2

Dear Editor and Reviewer,

First of all, we would like to thank you for the time you have spent reviewing our manuscript. We strongly appreciate the constructive comments and valuable feedback made. We have carefully addressed the reviewer's comments and suggestions. Below are our point-by-point responses to the comments in blue.

Thank you very much again for your review.

Author and Co-Authors

The study addresses the added value of convection-permitting modeling in extreme precipitation from regional to local scale, and the ability of convection-permitting model (HCLIM3) in reproducing orographic effects on precipitation in a topographically diverse country like Norway, by comparing it with those where convection is parameterized (HCLIM12). The evaluation considers both gridded datasets and in-situ observation (10 hourly rain-gauges and 192 daily rain-gauges), and provides a robust evaluation of the performance that HCLIM3 offers in the context of extreme precipitation modelling. A key contribution of the paper is its examination of the magnitude, frequency, seasonality and orographic effect of hourly and daily

extremes between HCLIM3 and HCLIM12 at both regional and local scales.

The results show that HCLIM3 provides added value over the HCLIM12 model

in most regions of Norway, particularly at the hourly scale. They highlight the

critical role of the convection-permitting regional climate model (HCLIM3) in

capturing the characteristics of extreme precipitation compared to HCLIM12.

This work holds great value for the application of regional climate models to

simulate and predict the severe meteorological hazards, particularly in the

context of localised extreme weather conditions. It provides critical

benchmarks the performance of convection-permitting model in local

extremes simulation.

Overall, this paper offers significant value and is suitable for publication in

HESS. The topic is of interest and fits the journal scope, but I have several

suggestions and comments before publication in HESS:

Major comments

1. Ten hourly rain gauges is a bit uncertain, so I suggest that you use it as

   additional remarks in the daily rain gauges section. You can delete it or

   remove some results related to hourly rain gauges to the Supplement,

   and indicate the uncertainty in the discussion.

Reply: Thank you for your valuable comment. We acknowledge that relying on

only ten hourly rain gauges introduces uncertainty. However, we emphasize

the importance of using both gridded and station data for analyzing hourly

extremes, as they provide complementary perspectives. The 9-year hourly gridded dataset corroborates the conclusions derived from the ten hourly rain gauges, showing that HCLIM12 underestimates the annual maximum 1-hour precipitation amount over Norway. Similarly, the ten hourly rain gauges consistently highlight this underestimation. Both the gridded dataset and rain gauge observations also underscore the added value of HCLIM3 compared to HCLIM12 for hourly extreme precipitation.

While we recognize the limitations of assessing HCLIM3's performance with this limited set of hourly rain gauges, we believe the data still offer valuable insights into the added value of convection-permitting regional climate models (CPRCMs). In response to your suggestion, we will revise the discussion and conclusions to explicitly address these uncertainties and clarify their implications. Additionally, we will move the results related to the 10 hourly rain gauges to the Supplement and include them as supplementary remarks in the rain gauge section, ensuring the main text remains focused and concise.

2. Conclusions should be drawn with caution, especially for hourly scale. Given that the length of the hourly gridded dataset is only nine years and that there are only ten hourly rain-gauges, it is therefore essential to exercise particular caution and awareness when considering the conclusions drawn from hourly in-situ observation.

Reply: Thanks for your valuable comment. We agree that the quality and availability of hourly observations are limited compared to daily data. While the 9-year gridded dataset is relatively short for robust statistical analysis, it provides comprehensive regional coverage, and we complemented it with in-situ observations spanning 20 years from ten stations. Despite these limitations, both datasets consistently show that HCLIM12 underestimates the hourly extreme precipitation, while HCLIM3 demonstrates clear improvements.

We acknowledge the uncertainties in hourly-scale analysis due to data constraints and will revise the discussion and conclusions to explicitly address these limitations and emphasize caution in interpreting the results.

3. The text uses a lot of acronyms for HCLIMs, but you don't define it. Please define this acronym at the first instance of its use.

Reply: Thanks for your valuable comment. We have added the definition of it in the revised manuscript: "HCLIMs indicates both HCLIM3 and HCLIM12."

4. Although this manuscript is well written, it should be edited further to ensure clarity for the reader. This should include attention to sentence structure, as well as minor spelling and grammatical errors.

Reply: Thanks for your valuable comment. We will revise the English language throughout the manuscript to enhance readability.

Minor comments

1. Figure 2, 6, 7: What do the dashed lines represent?

Reply: Thanks for your comment. The dashed lines represent the mean bias. We have added the explanation of the dashed lines in the revised manuscript.

2. Figure 10, 11, 12, 15: Replace the "Station ID" with "Name". Revise the corresponding text.

Reply: Thanks for your comment. We have updated the plots and corrected the text in the revised manuscript.

3. Figure 13-15: the unit is missing.

Reply: Thanks for your comment. We have added the unit of the plots in the Figures.

4. Figure 16, 17, 18: The title of the figure is unclear, please revise it.

Reply: Thanks for your comment. We have rewritten the titles of these figures in the revised manuscript. Please see below:

"Figure 16: Relationship between elevation and Rx1d (maximum 1-day precipitation) for (a) winter, (b) spring, (c) summer, and (d) autumn, as derived from SeNorge and HCLIMs (i.e., HCLIM3 and HCLIM12) across mainland Norwegian during the period of 1999-2018."

"Figure 17: Relationship between elevation and Rx1d (maximum 1-day precipitation) for (a) winter, (b) spring, (c) summer, and (d) autumn, based on

daily in-situ observation and HCLIMs (i.e., HCLIM3 and HCLIM12) across mainland Norwegian during the period of 1999-2018."

"Figure 18: Relationship between elevation and Rx1h (maximum 1-hour precipitation) for (a) winter, (b) spring, (c) summer, and (d) autumn, as derived from SeNorge2 and HCLIMs (i.e., HCLIM3 and HCLIM12) across mainland Norwegian during the period of 2010-2018."

5. Table 1: Unit of the "Elevation" is missing. Check and move to supplement. If it is possible, a corresponding Table for the detail information of daily rain-gauges is necessary.

Reply: Thanks for your comment. We have added the unit of the Elevation (m) and moved the table to supplement. We also added the information of daily rain-gauges in Table S2 in the supplement.

6. Line 200 and Figure 5, 9: Why statistic the frequency exceeding 10, 15 and 20 mm, how to define the threshold? Given the focus of your paper on extreme precipitation, it would be advisable to remove these results.

Reply: Thanks for your comment. Precipitation intensity of 20 mm/hour are considered rare extreme events that can trigger severe flooding. To assess these events, we calculate the frequency of precipitation exceeding 20 mm/hour, as well as smaller thresholds of 10 mm/hour and 15 mm/hour. Upon review, we found Figures 5, 9, and 12 redundant, as Figures 13, 14, and 15 already present the precipitation frequency data. Therefore, we will remove Figures 5, 9, and 12 to streamline the analysis.

7. Line 49-51: Please rewrite the sentence.

Reply: Thanks for your comment. We have rewritten the sentence as follows: "However, most previous research in this field has relied on coarse-resolution GCMs with grid sizes exceeding 100 km, which struggle to accurately simulate extreme precipitation events and their frequency due to the limitations of their coarser resolution".

8. Line 62: "improve the estimates of short-duration extremes". Please correct it.

Reply: Done.

9. Line 64-65: Replace "atmospheric deep convection" with "deep atmospheric convection".

Reply: Done.

10. Line 84: Replace "coarser-scale model" with "a coarser-scale model".

Reply: Done.

11. Line 87: Delete "were".

Reply: Done.

12. Line 128-135: Please use either CPRCM or CPRCMs consistently.

Reply: Thanks for your comment. We have uniformed them in the revised manuscript as below.

"Convection-permitting Climate Models (CPRCMs)"

13. Line 132-134: "The main objectives of this study are (1) enhance...; (2) assess...". Please revise it.

Reply: Thanks for your comment. We have revised it:

"The main objectives of this study are to: (1) enhance understanding of convection-permitting climate models by comparing their effectiveness in simulating extreme precipitation with that of regional climate models from regional to local scales, highlighting the added value of CPRCMs; (2) assess HCLIM3's capability in depicting orographic effects on seasonal extreme precipitation. This research explores whether the benefits provided by CPRCMs hold consistently in different regions driven by varying physical processes for precipitation."

14. Line 161-165: This contradicts the AR argument as ARs are always associated with extratropical cyclones.

Reply: Thanks for your comment. We will revise this part according to your comments.

15. Line 177, 178, 264: Replace "Norway mainland" with "Norwegian mainland".

Reply: Done. We will unify the term in the whole paper: Norwegian mainland.

16. Line 193: Please elucidate the rationale behind the decision to resample to HCLIM12 (~12 km). What are the distinguishing factors between resampling to HCLIM3 (~3 km) and the aforementioned approach?

Reply: Thanks for your comment. We have clarified this point in the revised manuscript.

Resampling coarse-resolution data (e.g., HCLIM12, 12 km) to finer resolution

can introduce artificial variability or spurious details, which not present in the original data, potentially leading to misleading conclusions. Conversely, resampling finer-resolution data to a coarser resolution reduces the influence of such artifacts by averaging out the variability. This approach aligns with methodology used by Lind et al. (2020) and Médus et al. (2022), who also remapped all data to a coarser grid when comparing the performance of HCLIM3 and HCLIM12. Lind et al. (2020) observed that the differences between HCLIM3 data remapped to the coarser native grid of HCLIM3 and the HCLIM12 grid were minimal. Importantly, they found that the improved performance of HCLIM3 persisted even after spatial aggregation, indicating that the model's enhanced resolution offered benefits that were preserved when viewed on a coarser grid. Please see the contexts in the revised manuscript as follow:

17.  Line 207-209: This is quite confusing. Please write it.

Reply: Thanks for your comment. We have rewritten it as: "The Generalized Extreme Value (GEV) distribution was used to estimate precipitation intensity for specified return periods (e.g., 5, 10, 20, and 50 years). This was done by fitting the GEV distribution to the cumulative distribution functions derived from the annual maximum precipitation intensities, including Rx1d and Rx1h, in the precipitation series from SeNorge, in-situ observation, HCLIM3 and HCLIM12."

18.  Line 224: Replace "relations" with "relationship".

Reply: Done.

19. Line 276: The sentence is not clear. Please rewrite it.

Reply: Thanks for your comment. We have rewritten the sentence as follows: "In the southern and southwestern regions, the annual Rx1d empirical distribution from both HCLIMs models closely align with those from SeNorge, making it difficult to distinguish which model performs better."

20. Line 385-386: The sentence is not clear. Please rewrite it.

Reply: Thanks for your comment. We have rewritten it as follows: " For example, the added value of HCLIM3 is shown at the regional scale in the middle-coastal region, but this advantage diminishes when analyzed at the local scale."

21. Line 386: Replace "in the middle-coastal" with "in the middle-coastal region".

Reply: Done.

22. Line 411-413: The sentence is not clear. Please rewrite it.

Reply: Thanks for your comment. We have rewritten it to be: "Based on station statistics for the mean annual Rx1d in Norway (Fig. 8), the boxplot shows that the mean annual Rx1d from HCLIM3 lines within the range of observed values. In contrast, HCLIM12 consistently underestimates Rx1d, with all its values falling below the observed minimum."

23. Line 431: Replace "that of HCLIM12" with "that from HCLIM12".

Reply: Done.

24. Line 481: "The CPRCM excels......". Check and rewrite it.

Reply: Thanks for your comment. We have rewritten it as follows: " The CPRCMs demonstrate better potential performance in reproducing Rx1h compared to RCMs, at both regional and station scales, with particularly improved accuracy at the localized scale."

25. Line 560-562: The sentence is not clear. Please correct it.

Reply: Thanks for your comment. We have rewritten the sentence as follow: "The comparison of HCLIM3 and HCLIM12 for seasonal Rx1d reveals that HCLIM3 provides a better representation overall although it exhibits a dry bias in the southwestern region. This dry bias may be attributed to the limitations of HCLIM3 in capturing unique precipitation mechanisms within this region."

26. Line 567-569: The sentence is not clear. Please rewrite it.

Reply: Thanks for your comment. We have rewritten it to be: " The impacts of station density on errors in gridded datasets was also highlighted by Gervais et al. (2014b), who identified low station density as a significant source of errors in such datasets."

27. Line 579: Replace "and found……" with "finding……".

Reply: Done.

28. Line 591: Replace "may attribute to" with "may be attribute to".

Reply: Done.

29. Line 594: "attributable to ……". Check and rewrite it.

Reply: Thanks for your comment. We have rewritten it:

"We observe an underestimation of extremes, including return levels and their temporal evolution, in HCLIM12 at the local scale, likely due to limitations in its convection-parameterization scheme."

30. Line 596: "especially at highly localized local scale". Check and rewrite it.

Reply: Thanks for your comment. We have rewritten it to be: "especially at highly localized scales".

31. Line 598: Replace "recognize" with "acknowledge".

Reply: Done.

32. Line 599: "The performance of RCM ~10 km in representing sub-daily rainfall was limited……". Please correct it.

Reply: Thanks for your comment. We have corrected it in the revised manuscript to be: "The performance of RCMs with a resolution of approximately 10 km in representing sub-daily rainfall is limited, as it has been shown to be challenging to capture sub-daily extreme rainfall, particularly in the southwestern United States."

33. Line 637: Replace the "shows " with "show".

Reply: Done.

34. Line 659: Delete "underestimation and".

Reply: Done.

35. Line 692: Replace the "HCLIM3 also have" with "HCLIM3 also has"

Reply: Done.

Lind, P., Belušić, D., Christensen, O. B., Dobler, A., Kjellström, E., Landgren, O., Lindstedt, D., Matte, D., Pedersen, R. A., Toivonen, E., and Wang, F.: Benefits and added value of convection-permitting climate modeling over Fenno-Scandinavia, Climate Dynamics, 55, 1893-1912, 10.1007/s00382-020-05359-3, 2020.

Médus, E., Thomassen, E. D., Belušić, D., Lind, P., Berg, P., Christensen, J. H., Christensen, O. B., Dobler, A., Kjellström, E., Olsson, J., and Yang, W.: Characteristics of precipitation extremes over the Nordic region: added value of convection-permitting modeling, Nat. Hazards Earth Syst. Sci., 22, 693-711, 10.5194/nhess-22-693-2022, 2022.

Thanks very much for your input, which helps us improve the quality and

clarity of our manuscript!

---

## Author Response (AR1)

**REPLY TO THE COMMENTS OF THE REFEREE**

Dear Editor and Reviewers,

First of all, we would like to thank you for the time you have spent reviewing our manuscript. We strongly appreciate the constructive comments and valuable feedback made. We have carefully addressed the reviewer's comments and suggestions. Below are our point-by-point responses to the comments in blue.

Thank you very much again for your review.

Author and Co-Authors

**COMMENTS FROM REVIEWER#1**

The paper presents an evaluation of extreme precipitation from two regional climate models; one is a regional model at 12km resolution, the other is a convection-permitting model at 3km resolution. The evaluation is based on different observational products: two grid products at 1day and 1hour temporal resolution, and a network of rain gauges (about 190 at daily resolution and 10 at hourly resolution). Seasonality and relation with elevation are also explored, at regional (by contrast with the gridded product) and local scale (by contrast with rain gages). The main findings indicate generally better performance for the HCLIM3 than the HCLIM12 model, more clearly at hourly resolution.

The study is of interest on the general topic of evaluation of extreme precipitation from convection-permitting models, which may help in better understanding how to use them in practical applications. In my opinion, it gives an incremental advancement in this field more than novelty, considering the regional scale (Norway) and the use of metrics commonly used in these kind of studies (Rx1d, Rx1h, return levels). It is well written, but I found difficult to get the main messages on the results because of the total length, number of figures and panels. The topic is of interest, and fitting the journal scopes, but I suggest a few major revisions and some minor before publication in HESS. My comments are listed below.

• Major 1. You used 8 regions. I wonder if less can be used, pooling together smaller ones. I say this based on two considerations: 1) extension is very different across the regions, and regional evaluation of models are then based on quite different number of grid points (for example, how many grid points for the two small regions in the south?) and number of daily rain gages (just 4-11-14 for three regions!); 2) it is difficult to follow the explanations and figures with comparisons on 8 regions, 2 models, 4 seasons, 2 durations ... and get a message on the results; maybe having less could help.

Reply: Thanks for your valuable comment! Based on your suggestions, we have revised the manuscript by merging the original eight regions into five broader regions: East (E), South (S), West (W), Middle (M), and North (N). This adjustment addresses the differences in spatial extent and the number of grid points and rain gauges across the regions. Additionally, it simplifies the interpretation of results, making comparisons across models, seasons, and durations more concise and easier to follow. We have updated all relevant figures, tables, and text throughout the manuscript accordingly. This revision has significantly improved the clarity and coherence of our analysis. • Major 2. Regional scale is here referred to the analysis using the gridded products; local scale is referred to the analysis based on rain gages. They show different results, but I wonder how much this is due to the use of different observation products. How is SeNorgemodels comparison sampled on the same rain gage points? Or, how is seNorge compared to rain gages? Differences you highlight in your text (e.g lines 337, 352, 384, etc) could be due to the use of a different observational benchmark. Moreover, when comparing gridded products and climate model, you compare values from a same size grid (12x12km), when comparing model and rain gages, the comparison is made on a 12x12km grid and a point measurement. I suggest to add further analysis comparing seNorge with rain gages, or extracting the comparison between seNorge vs models on the same locations of rain gages, or to add some considerations in the discussion.

Reply: We fully agree with your comment. We have analyzed and compared seNorge with rain gages by interpolating the seNorge to the same location of rain gauges. We have added it in the discussion (section 5.1) in the revised manuscript.

• Major 3. Biases (e.g. figure 2, 6) are shown in mm (absolute differences). Maybe relative differences (modelledobserved)/observed could be more meaningful considering that precipitation has a big range across the regions: a bias of 5 mm is different on a 30 mm or a 80 mm daily precipitation! My suggestion is to update maps and plots of biases with relative bias (%), and to revise description of results and comments on "magnitude" of bias based on this.

Reply: Thanks for your comment. We have updated all the plots with relative bias (%). In order to avoid the cancellation of positive and negative deviations when calculating the mean bias for each region, the absolute value of the percentage deviation is taken and averaged, as shown in the Fig. 2 (c), Fig. 3 (c) and Fig. 5 (c) in the revised manuscript.

**Minor comments**

title: I suggest hourly in place of sub-daily, considering that you just evaluate
1h

Reply: Thanks for the comment, the titles were changed for: "Enhanced Evaluation of hourly and Daily Extreme Precipitation in Norway from Convection-Permitting Models at Regional and Local Scales".

2. Line 120: these results on orographic effect "were based on the annual maxima"... yes. But also yours are based on Rx1d (see line 222-223). So ... why

do you highlight these about other studies at line 120, if then you do the same? I suggest to remove.

Reply: Thanks for your comment. We have revised it in the revised manuscript (Line 124-125), as shown below:

"It is worth noting that the potential added value of CPRCMs in representing orographic effects compared to RCMs has not been explored."

 Line 121-122. "The dependence on seasonality ... need the evaluation based on season". Of course! Maybe you wanted to tell something different here and I didn't get it. Please calrify/modify.

Reply: Thanks for your comment. We appreciate your feedback. What we intended to convey is that the performance of CPRCMs is influenced by seasonal variations, which necessitates evaluating the orographic effects of seasonal extremes. This is crucial for understanding how different seasons impact model performance and the resulting hydrological responses. We revised it in the revised manuscript (Line 125-126), as shown below:

"Moreover, the performance of CPRCMs varies with seasons, which underscores the need to explore the orographic effects on seasonal extremes."

Line 195. Not clear: you say here you averaged the indices in the region ... then in caption of figure 2 you write the bias is calculated at each grid point (this makes sense to me). So, when do you use the averaged indices?

Reply: Thanks for your comment. To clarify, the bias is initially calculated at each grid point within the region. After calculating the bias for each grid point, we then average these biases across all grid points within each region to obtain a regional average bias. This approach ensures that the regional bias is representative of all grid points in the region.

We have revised it more clearly in the revised manuscript (Line 205-206), as shown below:

"For the SeNorge and SeNorge2 based assessments, the extreme indices are first calculated at the grid-point level and then the regional averages are computed."

4. Line 206. Specify somewhere in the section that this is done for daily data (both gridded and rain gages), while just on 10 rain gages at 1h duration. More importantly, later in the text (lines 367, 372,..) you mention uncertainty ... and it is never explained before in the paper how it is evaluated. Add it in the methodology.

**Reply: Thanks for your comment.**

(i)We have revised it in the revised manuscript (Line 214-217), as shown below:

"The Generalized Extreme Value (GEV) distribution was used to estimate precipitation intensity for specific return periods (e.g., 5, 10, 20, and 50 years). The return levels were calculated by fitting the annual maximum discharge derived from observed and simulated daily data (both gridded and rain gauges), and hourly data (only 10 rain gauges), to GEV distribution."

(ii) Uncertainty is assessed by evaluating the variability among the stations. This is represented in boxplots (Fig. S1, in supplementary), where the long whiskers indicate the range of variability across all stations within each region, illustrating the uncertainty in the return level estimates.

We have modified the "uncertainty" to "variations", and rewritten it more clearly in the revised manuscript (Line 348-350), as shown below:

"In addition, Fig. S1 shows the range of the return levels for all stations in the corresponding region, and HCLIM3 introduces larger variations in the western and southern regions compared with HCLIM12, as indicated by the wider whiskers."

5. Figure 2 (same for figures 6 and 7). The caption mentions absolute bias but describes a relative bias calculation. Please correct.

Reply: Thanks for your comments. We have corrected it to percentage bias (simulations minus observations, divided by observations) in the revised manuscript, as shown below:

"Figure 2: (a) The annual Rx1d of SeNorge, and the percentage bias of Rx1d from HCLIM3 and HCLIM12 to SeNorge during 1999-2018; (b) density plot of the percentage bias distribution for annual Rx1d from HCLIM3 and HCLIM12 compared to SeNorge for Rx1d during 1999-2018 (The dashed lines represent the mean bias); (c) the absolute percentage bias of annual and seasonal Rx1d from HCLIM3 and HCLIM12 to SeNorge for five regions. The bias is first calculated at the grid-point level, and then regional averages are computed. For (a) and (b), the percentage bias is equal to model simulations minus observations, divided by observations. For (c), the absolute percentage bias is calculated as the absolute difference between simulations and observations, divided by observations."

"Figure 3: (a) The annual Rx1h of SeNorge2, and the percentage bias of Rx1h from HCLIM3 and HCLIM12 to SeNorge2 during 2010-2018; (b) density plot of percentage bias for annual Rx1h distribution from HCLIM3 and HCLIM12 compared to SeNorge2 during 2010-2018 (The dashed lines represent the mean bias); (c) the absolute percentage bias of seasonal Rx1h from HCLIM3 and HCLIM12 to SeNorge2 for five regions. For (a) and (b), the percentage bias is equal to model simulations minus observations, divided by

observations. For (c), the absolute percentage bias is calculated as the absolute difference between simulations and observations, divided by observations."

"Figure 5: (a) The annual Rx1d of in-situ observation, and the percentage bias of Rx1d from HCLIM3 and HCLIM12 to in-situ observation during 1999-2018 over 194 stations; (b) density distribution of percentage bias for annual Rx1d between HCLIMs and observations from 194 stations during 1999-2018 (The dashed lines represent the mean bias); (c) the absolute percentage bias of seasonal Rx1d between HCLIMs and observations across the five regions. For (a) and (b), the percentage bias is equal to model simulations minus observations, divided by observations. For (c), the absolute percentage bias is calculated as the absolute difference between simulations and observations, divided by observations."

6. Figure 2c (same for figures 6 and 7). I suggest to add another row with the regional bias for annual Rx1d, not just seasonal, in order to have a synthesis of what is shown in the maps in panel a.

Reply: We fully agree with your comment. We have added the row with regional bias for annual Rx1d in the revised manuscript, as shown in the updated Fig. 2, Fig. 3 and Fig. 5.  Figure 4. Same y-axis limits could help in comparing the bias... and maybe you can revise the description of results better considering the different magnitude of the bias (example at lines 290-291).

Reply: We agree. We have updated the plot with same y-axis limits in Fig. 4 of the revised manuscript.

8. Figure 4. I can't understand why Northern-Coastal has so big bias for HCLIM3, considering that figure 3 shows a tendency of underestimation of the empirical distribution, similar to Northern-Inland, for which you find big underestimation of return levels. And for the Norther-Inland, why underestimation of return levels is bigger for HCLIM3, while the distribution in figure 3 show more underestimation for HCLIM12? ... please check the correctness of the results.

Reply: Thanks for your comment and for pointing out the discrepancies. Upon review, we recalculated the return levels for all regions and found that the calculations for the Northern-Coastal and Northern-Inland regions were indeed incorrect. As a result, we have corrected these errors and updated the results and corresponding plot. We apologize for the confusion. Thank you again for bringing this to our attention. The plot for eight regions is shown below:

Figure 4: The bias of extreme annual Rx1d exceeding the 5-year to 50-year over eight regions between SeNorge and HCLIMs (i.e., HCLIM3 and HCLIM12).

---

## Author Response (AR2)

**REPLY TO THE COMMENTS OF THE REFEREE**

**Dear Editor and Reviewers,**

**First of all, we would like to thank you for the time you have spent reviewing our manuscript. We appreciate all comments and valuable feedback made. Below are our point-by-point responses to the comments in blue.**

**Thank you very much again for your review.**

12. Line 494. "Significant"? Based on a specific test? Maybe "relevant"…

Reply: Thanks. We have deleted the significant in the revised manuscript (Line 440).

I cannot find the revision in Line440.

Reply: Thanks for your comments. We have revised the text and the paragraph in this section. Instead of changing the word, we decided to remove the entire sentence in the revision. This is why it is not visible in the manuscript. Our previous reference to "Line 440" was misleading, and we apologize for the confusion.

Line231-233: To eliminate the impact of unit (Rx1h and Rx1d), the slope is transferred to relative slope with respect to the average value of extreme

precipitation, expressed as percentage precipitation (%) per kilometer of elevation.

Please change 'transferred' to 'converted'.

Reply: Done. (Line 234).

'with respect to the average value of extreme precipitation': please specify the average value of extreme precipitation for daily and hourly. It is not clear what reference values you used here for normalising the percentage change.

Reply: Thanks for your comments. We have clarified this in the revised manuscript (Line 234-237), as shown below:

"To eliminate the impact of unit (Rx1h and Rx1d), the slope is converted to a relative slope with respect to the average value of extreme precipitation, expressed as percentage precipitation (%) per kilometer of elevation. This is done by dividing the mean extreme precipitation value for the entire study region computed separately for daily and hourly extremes."